# Association mapping from sequencing reads using *k*-mers

**Atif Rahman[1†]\*, Ingileif Hallgrímsdóttir[2‡], Michael Eisen[3,4], Lior Pachter[1,3,5§]\***

[1]Department of Electrical Engineering and Computer Sciences, University of California, Berkeley, Berkeley, United States; [2]Department of Statistics, University of California, Berkeley, Berkeley, United States; [3]Department of Molecular and Cell Biology, University of California, Berkeley, Berkeley, United States; [4]Howard Hughes Medical Institute, University of California, Berkeley, Berkeley, United States; [5]Department of Mathematics, University of California, Berkeley, Berkeley, United States

**\*For correspondence:**
atif@cse.buet.ac.bd (AR);
lpachter@caltech.edu (LP)

**Present address:** [†]Department of Computer Science and Engineering, Bangladesh University of Engineering and Technology, Dhaka, Bangladesh; [‡]Amgen, California, United States; [§]Departments of Biology and Computing & Mathematical Sciences, California Institute of Technology, Pasadena, United States

**Competing interests:** The authors declare that no competing interests exist.

**Abstract** Genome wide association studies (GWAS) rely on microarrays, or more recently mapping of sequencing reads, to genotype individuals. The reliance on prior sequencing of a reference genome limits the scope of association studies, and also precludes mapping associations outside of the reference. We present an alignment free method for association studies of categorical phenotypes based on counting *k*-mers in whole-genome sequencing reads, testing for associations directly between *k*-mers and the trait of interest, and local assembly of the statistically significant *k*-mers to identify sequence differences. An analysis of the 1000 genomes data show that sequences identified by our method largely agree with results obtained using the standard approach. However, unlike standard GWAS, our method identifies associations with structural variations and sites not present in the reference genome. We also demonstrate that population stratification can be inferred from *k*-mers. Finally, application to an *E.coli* dataset on ampicillin resistance validates the approach.
DOI: https://doi.org/10.7554/eLife.32920.001

## Introduction

Association mapping refers to the linking of genotypes to phenotypes. Most often this is done using a genome-wide association study (GWAS) with single nucleotide polymorphisms (SNPs). Individuals are genotyped at a set of known SNP locations using a SNP array. Then each SNP is tested for statistically significant association with the phenotype. In recent years thousands of genome-wide association studies have been performed and regions associated with traits and diseases have been located.

However, this approach has a number of limitations. First, designing SNP arrays requires knowledge about the genome of the organism and where the SNPs are located in the genome. This makes it hard to apply to study organisms other than human. Even the human reference genome was shown to be incomplete (*Altemose et al., 2014*) and association mapping to regions not in the reference is difficult. Second, structural variations such as insertion-deletions (indels) and copy number variations are usually ignored in these studies. Despite the many GWA studies that have been performed a significant amount of heritability is yet to be explained. This is known as the 'missing heritability' problem (*Zuk et al., 2012*). A hypothesis is some of the missing heritability is due to structural variations. Third, the phenotype might be caused by rare variants which are not on the SNP chip. In last two cases, follow up work is required to find the causal variant even if association is detected in the GWAS.

Some of these limitations can be overcome by utilizing high throughput sequencing data. As sequencing gets cheaper association mapping using next generation sequencing is becoming feasible. The current approach to doing this is to map all the reads to a reference genome followed by variant calling. Then these variants can be tested for association. But this again requires a reference genome and it may induce biases in variant calling and regions not in the reference genome will not be included in the study. Moreover, sequencing errors make genotype calling difficult when sequencing depth is low (*Nielsen et al., 2012*) and in repetitive regions. Methods have been proposed to do population genetics analyses that avoid the genotype calling step (*Fumagalli et al., 2013*, *2014*) but these methods still require reads to be aligned to a reference genome. An alternate approach is simultaneous de novo assembly and genotyping using a tool such as Cortex (*Iqbal et al., 2012*) but this is not suited to large number of individuals as simultaneous assembly and variant calling need loading all $k$-mers from all samples into memory requiring large amount of it. The alternative approach of loading a subset of samples to process at a time would require subsequent alignment of sequences. Neither of these approaches is trivially parallelizable.

In the past, alignment free methods have been developed for a number of problems including transcript abundance estimation (*Patro et al., 2014*), sequence comparison (*Song et al., 2014*), phylogeny estimation (*Haubold, 2014*), etc. (*Nordström et al., 2013*) introduced a pipeline called needle in the $k$-stack (NIKS) for mutation identification by comparison of sequencing data from two strains using $k$-mers. (*Sheppard et al., 2013*) presented a method for association mapping in bacterial genomes using $k$-mers. More recently, (*Earle et al., 2016*) proposed a method for mapping associations to lineages when associations can not be accurately mapped to loci and (*Lees et al., 2016*) showed that use of variable length $k$-mers leads to an increase in power.

However, these methods for association mapping in bacterial genomes use only the presence and absence of $k$-mers and ignore the actual counts. This prevents association mapping to copy number variations (CNVs). Moreover, tests based on $k$-mer counts are likely to have more power, making detection of association with smaller number of samples possible (*Appendix 1—figures 2* and *3*). Here we present an alignment free method for association mapping to categorical phenotypes. It is based on counting $k$-mers and identifying $k$-mers associated with the phenotype. The overlapping $k$-mers found are then assembled to obtain sequences corresponding to associated regions. Our method is applicable to association studies in organisms with no or incomplete reference genome. Even if a reference genome is available, this method has the advantage of avoiding aligning and genotype calling thus allowing association mapping to many types of variants using the same pipeline and to regions not in the reference.

In contrast to the approach in *Iqbal et al. (2012)*, in our method, $k$-mers are initially tested for association independently of other $k$-mers allowing us to load only a subset of $k$-mers using lexicographic ordering. However, their approach can utilize information from the reference genome if one is available whereas we currently make use of the reference genome only after sequences associated have been obtained to determine the type of associated variant. A future direction may be to utilize this information earlier in the pipeline.

We have implemented our method in a software called 'hitting associations with $k$-mers' (HAWK). Experiments with simulated and real data demonstrate the promises of this approach. To test our approach in a setting not confounded by population structure, we apply our method to analyze whole genome sequencing data from three populations in the 1000 genomes project treating population identity as the trait of interest.

In a pairwise comparison of the Toscani in Italia (TSI) and the Yoruba in Ibadan, Nigeria (YRI) populations we find that sequences identified by our method largely agree with results obtained using standard GWAS based on variant calling from mapped reads (Figure 2). Agreement with sites found using read alignment and genotype calling indicate that $k$-mer based association mapping will be applicable to mapping associations to diseases and traits.

We also analyze data from the Bengali from Bangladesh (BEB) population to explore possible genetic basis of high rate of mortality due to cardiovascular diseases (CVD) among South Asians and find significant differences in frequencies of a number of non-synonymous variants in genes linked to CVDs between BEB and TSI samples, including the site rs1042034, which has been associated with higher risk of CVDs previously, and the nearby rs676210 in the *Apolipoprotein B (ApoB)* gene.

We then demonstrate that population structure can be inferred from $k$-mer data from whole genome sequencing reads and discuss how population stratification and other confounders can be

accounted for. Finally, we apply our method to *E. coli* data set on ampicillin resistance and find hits to the *β-lactamase TEM (blaTEM)* gene, the presence of which is known to confer ampicillin resistance, validating our overall approach.

## Materials and methods

### Association mapping with *k*-mers

We present a method for finding regions associated with a categorical trait using sequencing reads without mapping reads to reference genomes. The workflow is illustrated in *Figure 1*. Given whole genome sequencing reads from case and control samples, we count *k*-mers appearing in each sample. We assume the counts are Poisson distributed and test *k*-mers for statistically significant association with case or control using likelihood ratio test for nested models (see Appendix 1 for details). Population structure is then inferred from *k*-mer data and used to adjust p-values. The differences in *k*-mer counts may be due to single nucleotide polymorphisms (SNPs), insertion-deletions (indels) and copy number variations. The *k*-mers are then assembled to obtain sequences corresponding to each region.

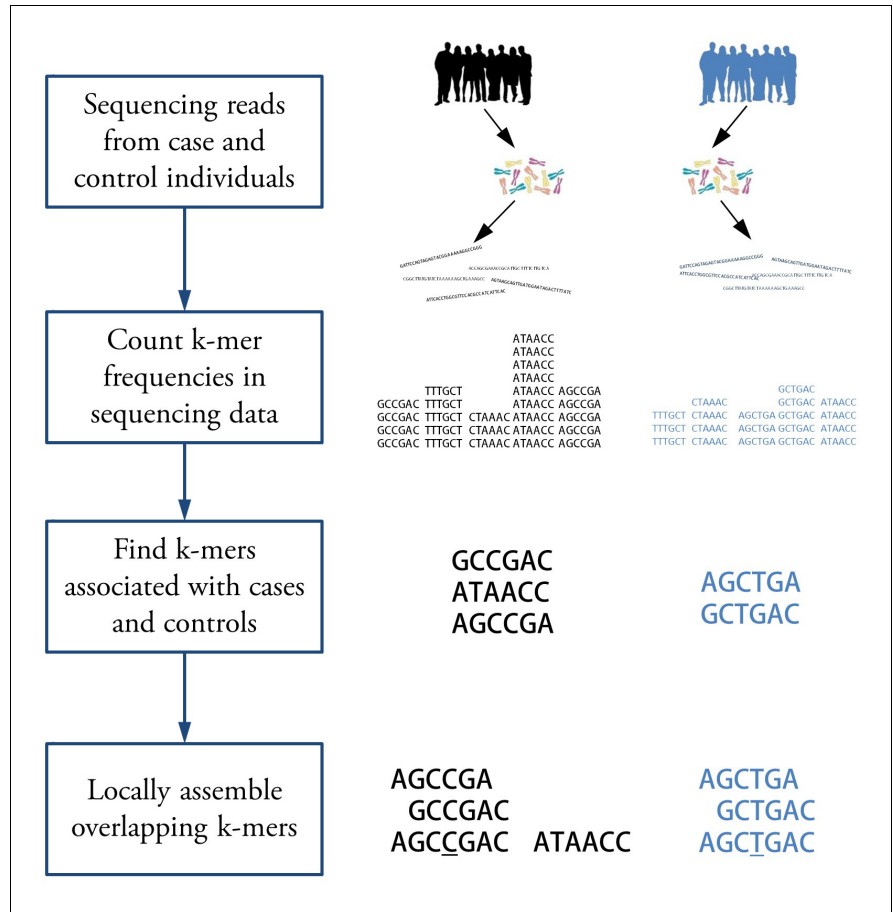

**Figure 1.** Workflow for association mapping using *k*-mers. The Hawk pipeline starts with sequencing reads from two sets of samples. The first step is to count *k*-mers in reads from each sample. Then *k*-mers with significantly different counts in two sets are detected. Finally, overlapping *k*-mers are assembled into sequences to get a sequence, shown side by side, for each associated locus. The sequences may correspond to a SNP (underlined) in which case corresponding sequence may be detected in the other group. This may not be the case for other kinds of variations such as copy number variation.

DOI: https://doi.org/10.7554/eLife.32920.002

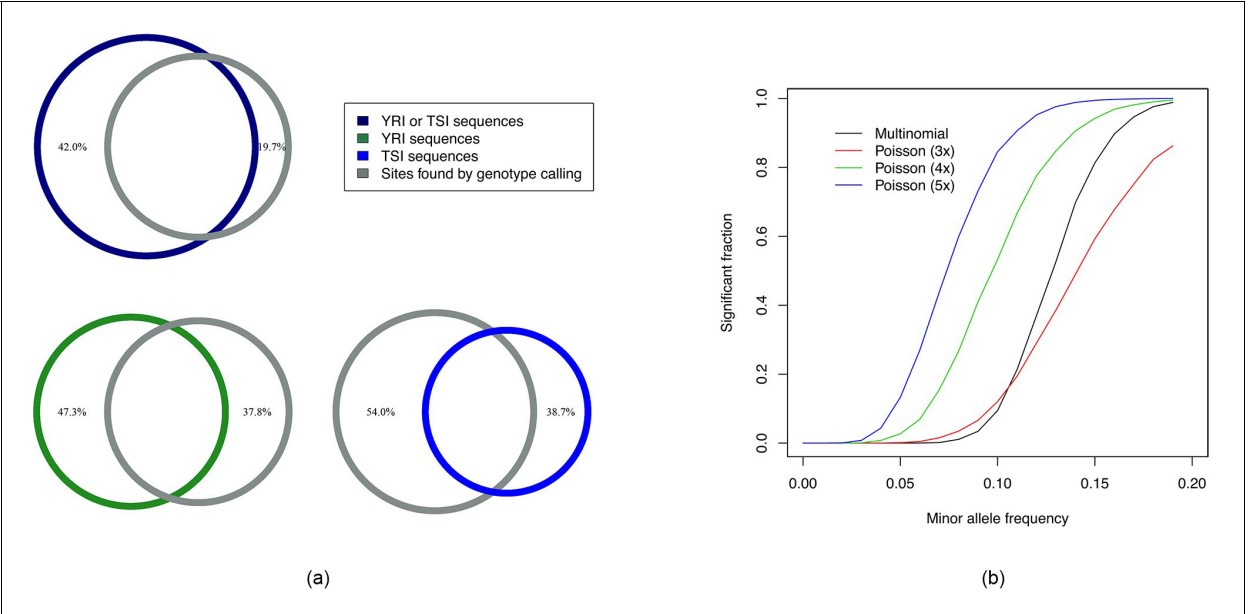

**Figure 2.** Intersection analysis and comparison of powers of tests. (a) Venn diagrams showing intersections among sequences obtained using HAWK and significant sites found by genotype calling. The percentage values shown are fractions of the sites found using one method not covered by those found by the other method. 80.3% of the sites overlapped with some sequence. Around 42% of sequences do not overlap with any such site which can be explained by more types of variants found by HAWK as well as more power of the test using Poisson compared to Multinomial distribution. (b) Fraction of runs found significant (after Bonferroni correction) by tests against minor allele frequency of the case samples (with that of the controls fixed at 0) are shown. The curves labeled multinomial and Poisson correspond to likelihood ratio test using multinomial distribution and Poisson distributions with different $k$-mer coverage.

DOI: https://doi.org/10.7554/eLife.32920.003

## Counting $k$-mers

The first step in our method for association mapping from sequencing reads using $k$-mers is to count $k$-mers in sequencing reads from all samples. To count $k$-mers we use the multi-threaded hash based tool JELLYFISH (*Marçais et al., 2011*). However, the pipeline can be modified to work with any $k$-mer counting tool. We use $k$-mers of length 31 which is the longest $k$-mer length which can be efficiently represented on 64-bit machines using JELLYFISH and filter out $k$-mers that appear once in samples before testing for associations for computational and memory efficiency as they are likely from sequencing errors.

## Finding significant $k$-mers

Then for each $k$-mer we test whether that $k$-mer appears significantly more times in case or control datasets compared to the other using a likelihood ratio test for nested models (*Wilks, 1938*). Suppose, the reads are of length $l$, then we observe a $k$-mer if a read starts in one of $l - k + 1$ positions at the start of the $k$-mer in the genome or preceding it. So, the count of a $k$-mer equals the number of reads that start in $l - k + 1$ positions, the probability of which is small for most $k$-mers as genomes tend to be much longer compared to that segment. That combined with the large number of reads in second generation sequencing motivates us to assume Poisson distributions which allows us to compute p-values quickly compared to negative binomial distributions used to model RNA-seq data (*Robinson and Smyth, 2008*; *van de Geijn et al., 2015*). Furthermore, we observe that for large number of $k$-mers in a whole genome sequencing experiment and typical counts of a single $k$-mer, p-values calculated assuming Poisson distributions are not notably lower than those obtained assuming negative binomial distributions (*Appendix 1—figure 1*).

Suppose, a particular $k$-mer appears $K_1$ times in cases and $K_2$ times in controls, and $N_1$ and $N_2$ are the total number of $k$-mers in cases and controls respectively. The $k$-mer counts are assumed to be Poisson distributed with rates $\theta_1$ and $\theta_2$ in cases and controls. The null hypothesis is $H_0{:}\theta_1 = \theta_2 = \theta$

and the alternate hypothesis is $H_1 : \theta_1 \neq \theta_2$. The likelihoods under the alternate and the null are given by (see Appendix 1 for details)

$$L(\theta_1, \theta_2) = \frac{e^{-\theta_1 N_1}(\theta_1 N_1)^{K_1}}{K_1!} \frac{e^{-\theta_2 N_2}(\theta_2 N_2)^{K_2}}{K_2!}$$

and

$$L(\theta) = \frac{e^{-\theta N_1}(\theta N_1)^{K_1}}{K_1!} \frac{e^{-\theta N_2}(\theta N_2)^{K_2}}{K_2!}.$$

Since the null model is a special case of the alternate model, $2ln\Lambda$ is approximately chi-squared distributed with one degree of freedom where $\Lambda$ is the likelihood ratio. We get a p-value for each $k$-mer using the approximate $\chi^2$ distribution of the likelihood ratio and perform Bonferroni corrections to account for multiple testing. We use the conservative Bonferroni correction as deviations from Poisson distributions are possible.

Our approach may be extended to quantitative phenotypes, by regressing phenotype values against $k$-mer counts to test whether $k$-mer counts are predictive of the phenotype.

## Detecting population structure

To detect population structure in the data, we randomly choose one-thousandth of the $k$-mers present between 1% and 99% of the samples and construct a binary matrix $B = \{b_{ij}\}$ where $b_{ij} = 1$ if the $j$-th $k$-mer is present in the $i$-th individual and 0 otherwise. We then perform principal components analysis (PCA) on the matrix which has been widely used to uncover population structure in genotype data (*Patterson et al., 2006*; *Price et al., 2006*). We have modified the EIGENSTRAT software (*Patterson et al., 2006*) to run PCA on $B$ and as in EIGENSTRAT, we normalize values in the $j$-th column using

$$m_{ij} = \frac{b_{ij} - \mu_j}{\sqrt{p_j(1 - p_j)}}$$

so that columns have approximately the same variance. Here $\mu_j$ is the mean of the $j$-th column and $p_j$ is the allele frequency estimated from the fraction of samples with the corresponding $k$-mer using the formula $p_j = 1 - \sqrt{1 - \mu_j}$ for diploids and $p_j = \mu_j$ for haploid organisms.

## Correcting for population stratification and other confounders

Population stratification is a known confounder in association studies. In association mapping from sequencing reads other possible confounding factors include variations in sequencing depth and batch effects. To correct for confounders in association mapping involving a categorical phenotype, for the $k$-mers found significant in the Poisson distribution based likelihood ratio test, we fit a logistic regression model on the phenotype against potential confounders and $k$-mer counts normalized using total number of $k$-mers in the sample. By default we include first two principal components obtained in the previous step and total number of $k$-mers in sequencing reads from each individual to account for population structure and varying sequencing depth respectively but other potential confounders may also be included as needed.

We then quantify the additional goodness of fit provided by each $k$-mer after the confounding factors and use R script to obtain an ANOVA p-value using a $\chi^2$-test with likelihood ratio and apply Bonferroni threshold established earlier. That is a logistic regression model is fitted against the confounders and probability of responses are used to compute likelihood. Similarly, another logistic regression model is fitted against the confounders as well as $k$-mer counts and likelihood under this model is computed. Since the former model is a special case of the latter, negative logarithm of the likelihood ratio is asymptotically $\chi^2$ distributed with one degree of freedom which is then used to calculate a p-value. For quantitative phenotypes linear regression may be used instead of logistic regression.

We performed simulations to compare powers of Poisson distribution based likelihood ratio test and logistic regression based tests to detect association for different coverages and varying number

**Table 1.** Known variants in YRI-TSI comparison.

*Table 1* shows p-values of sequences computed using likelihood ratio test at some well known sites of variation between populations. The (%) values denote fraction of individuals in the sample with the allele present. The p-values and % values are averaged over $k$-mers constituting the associated sequences.

| Gene | SNP id | Description | Allele | p-value | %YRI | %TSI |
|------|--------|-------------|--------|---------|------|------|
| ACKR1 | rs2814778 | Duffy antigen | C | $9.72 \times 10^{-114}$ | 84.39% | 1.78% |
| SLC24A5 | rs1426654 | Skin pigmentation | G | $8.45 \times 10^{-144}$ | 87.39% | 1.02% |
| SLC45A2 | rs16891982 | Skin/hair color | C | $1.89 \times 10^{-122}$ | 92.18% | 4.67% |
| G6PD | rs1050829 | G6PD deficiency | C | $1.53 \times 10^{-29}$ | 24.92% | 1.02% |
| G6PD | rs1050828 | G6PD deficiency | T | $5.83 \times 10^{-25}$ | 18.32% | 0.00% |

DOI: https://doi.org/10.7554/eLife.32920.004

of case individuals having one and two copies of the allele with number of copies in control individuals fixed at zero and one copy respectively. The results are shown in *Appendix 1—figure 4*. We observe that logistic regression based tests have less power compared to Poisson distribution based test. We also note that logistic regression based test using only presence and absence of $k$-mer has similar power as the one using $k$-mer counts while detecting one copy of an allele against zero copies but it is unable to detect association if cases have two copies of an allele against one copy in controls. We leave designing tests modeling stochasticity in counts incorporating confounders as well as extending our approach to quantitative phenotypes as future work.

## Merging $k$-mers

We then take $k$-mers associated with cases and controls and locally assemble overlapping $k$-mers to get a sequence for each differential site using the assembler ABySS (*Simpson et al., 2009*). The goal of this step is to have a sequences for each associated locus instead of having multiple $k$-mers from it. ABySS was used as the assemblies it generated were found to cover more of the sequences to be assembled compared to other assemblers (*Rahman and Pachter, 2013*). We construct the de Bruijn graph using hash length of 25 to be robust to lack of detection of some 31-mers without creating many ambiguous paths in the de Bruijn graph and retain assembled sequences of length at least 49 which is the length formed by 25-mers overlapping with a SNP site on either side. It is also possible to merge $k$-mers and pair corresponding sequences from cases and controls using the NIKS pipeline (see [*Nordström et al., 2013*] for details). However, we find that this is time consuming when we have many significant $k$-mers. Moreover, when number of cases and controls are not very high we do not have enough power to get both of the sequences to be paired and as such pairing is not possible.

## Implementation

Our method is implemented in a tool called 'hitting associations with $k$-mers' (HAWK) using C++. To speed up the computation we use a multi-threaded implementation. In addition, it is not possible to load all the $k$-mers into memory at the same time for large genomes. So, we sort the $k$-mers lexicographically and load them into memory in batches. To make the sorting faster JELLYFISH has been modified to output integer representation of $k$-mers instead of the $k$-mer strings. In future the sorting step may be avoided by utilizing the internal ordering of JELLYFISH or other tools for $k$-mer counting. The principal components analysis (PCA) is performed using a modified version of the widely used EIGENSTRAT software (*Patterson et al., 2006*). The logistic regression model fitting and p-value computation is done using scripts written in R and we are presently exploring ways to speed up the computation. The implementation is available at http://atifrahman.github.io/HAWK/ (copy archived at https://github.com/elifesciences-publications/HAWK).

## Downstream analysis

The sequences obtained by merging overlapping $k$-mers can then be analyzed by aligning to a reference if one is available or by running BLAST (*Altschul et al., 1990*) to check for hits to related

organisms. The intersection results in this paper were obtained by mapping them to the human reference genome version GRCh37 using Bowtie2 (*Langmead and Salzberg, 2012*) to be consistent with co-ordinates of genotypes called by 1000 genomes project. The breakdown analysis was performed by first mapping to the version of the human reference genome at the UCSC Human Genome Browser, hg38 and then running BLAST on some of the ones that did not map. Specific loci of interest were checked by aligning them to RefSeq mRNAs using Bowtie2 and on the UCSC Human Genome Browser by running BLAT (*Kent, 2002*).

## Results

### Verification with simulated data

The implementation was tested by simulating reads from the genome of an *Escherichia coli* strain. We introduced different types mutations - single nucleotide changes, short indels (less than 10 bp) and long indels (between 100 bp and 1000 bp) into the genome. Then wgsim of SAM tools (*Li et al., 2009*) was used to first generate two sets of genomes by introducing additional random mutations (both substitutions and indels) into the original and the modified genomes and then simulate reads with sequencing error rate of 1% and other default parameters of wgsim. The HAWK pipeline was then run on these two sets of sequencing reads. The fraction of mutations covered by resulting sequences are shown in *Appendix 1—figure 5* for varying numbers of case and control samples and different types of mutations. The results are consistent with calculation of power to detect $k$-mers for varying total $k$-mer coverage (*Appendix 1—figure 2*) with slightly lower values expected due to sequencing errors and conditions imposed during assembly.

### Verification with 1000 genomes data

To analyze the performance of the method on real data we used sequencing reads from the 1000 genomes project (*Abecasis et al., 2012*). The population identities were used as the phenotype of interest circumventing the need for correction of population structure. For verification, we used

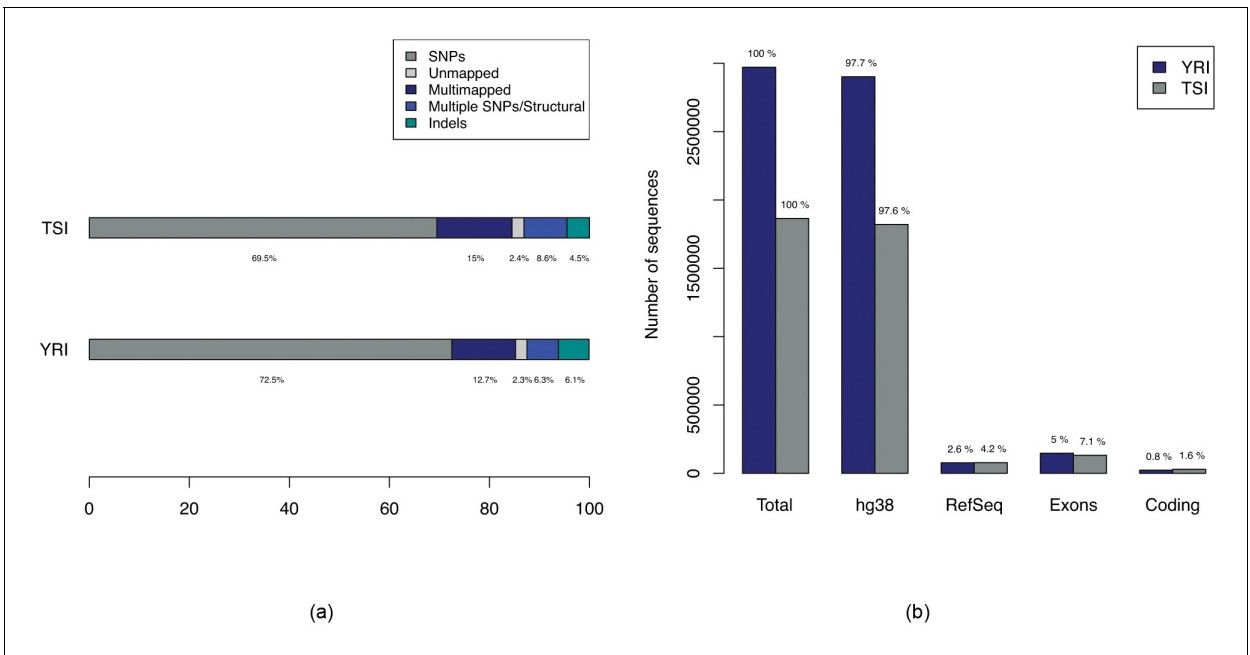

**Figure 3.** Breakdown of types of variations in comparison of YRI-TSI. (**a**) Bars showing breakdown of 2,970,929 and 1,865,285 sequences enriched for in YRI and TSI samples respectively. The 'Multiple SNPs/Structural' entries correspond to sequences of length greater than 61, the maximum length of a sequence due to a single SNP with $k$-mer size of 31 and 'SNPs' correspond to sequences of maximum length of 61. (**b**) Numbers of sequences with alignments to hg38, RefSeq mRNAs and Ensembl exons and coding regions.
DOI: https://doi.org/10.7554/eLife.32920.005

**Table 2.** Summary of sequences not in the human reference genome.
*Table 2* shows summary of sequences associated with different populations that did not map to the human reference genome (hg38) or to the Epstein-Barr virus genome.

| Population | Population compared to | Total no. sequences | No. sequences with length≥1000bp | Total length in sequences with length≥1000 bp | No. sequences with length≥200bp | Total length in sequences with length≥200bp |
|---|---|---|---|---|---|---|
| YRI | TSI | 94,795 | 41 | 59,956 | 478 | 225,426 |
| TSI | YRI | 66,051 | 10 | 13,896 | 184 | 77,383 |
| BEB | TSI | 19,584 | 3 | 3835 | 75 | 33,954 |
| TSI | BEB | 18,508 | 2 | 2105 | 81 | 28,134 |

DOI: https://doi.org/10.7554/eLife.32920.006

sequencing reads from 87 YRI individuals and 98 TSI individuals for which both sequencing reads and genotype calls were available at the time analysis was performed. The genotype calling was performed using the same set of reads we used to perform association mapping.

The analysis using $k$-mers revealed 2,970,929 sequences enriched for in the YRI population as compared to the TSI population and 1,865,285 sequences enriched for in TSI samples. QQ plot of the p-values obtained is shown in *Appendix 1—figure 6(a)*. Although we observe large deviations from the diagonal line, this is partly due to large number of $k$-mers for which the null is not true and cumulative distribution of the p-values, shown in *Appendix 1—figure 6(b)*, reveals that majority of the p-values are not significantly small.

To compare the results with the standard approach of mapping reads and calling variants, we also performed similar analysis with genotype calls available from the 1000 genomes project. VCFtools (*Danecek et al., 2011*) was used to obtain number of individuals with 0, 1 and 2 copies of one of the alleles for each SNP site. Each site was then tested to check whether the allele frequencies are significantly different in two samples using likelihood ratio test for nested models for multinomial distribution (details in Appendix 1). We found that 2,658,964 out of the 39,706,715 sites had allele frequencies that are significantly different.

*Figure 2a* shows the extent of overlap among these discarding the sequences that did not map to the reference. We used to BEDtools (*Quinlan and Hall, 2010*) to determine the number sites that were within an interval covered by at least one sequence found by assembling $k$-mers. We find that 80.3% (2,135,415 out of 2,658,964) of the significant sites was covered by some sequence found using Hawk while 19.7% was not as shown in the Venn diagram on the top left. Approximately 95.2% of the sites was covered by at least one $k$-mer.

We also observe that around 42% of sequences found using $k$-mers do not cover any sites found significant using genotype calling. While up to 20% of them correspond to regions for which we did not have genotype calls (chromosome Y, mitochondrial DNA and small contigs), repetitive regions where genotype calling is difficult and structural variations, many of the remaining sequences are possibly due to more power of the test based on counts than the one using only number of copies of an allele. We performed Monte Carlo simulations to determine powers of the two tests. *Figure 2 (b)* shows the fraction of trials that passed the p-value threshold after Bonferroni correction as the allele frequencies in cases were increased keeping the allele frequencies of control fixed at 0.

This is consistent with greater fraction of sequences in YRI (47.3% shown in bottom left of *Figure 2 (a)*) not covering sites obtained by genotyping compared to TSI (38.7% shown in bottom right of *Figure 2(a)*) as some low frequency variations in African populations were lost in other populations due to population bottleneck during the migration out of Africa. However, some false positives may result due to discrepancies in sequencing depth of the samples and sequencing biases. We provide scripts to lookup number of individuals with constituent $k$-mers to help investigate sites found using Poisson distribution based likelihood ratio test only. *Table 1* shows example p-values of some of the well known sites of variation between African and European populations as well as fraction of individuals in each group with the variant looked up using such scripts.

## HAWK maps associations to multiple variant types

HAWK enables mapping associations to different types of variants using the same pipeline. *Figure 3 (a)* shows breakdown of types of variants found associated with YRI and TSI populations. The 'Multiple SNPs/Structural' entries correspond to sequences of length greater than 61 (the maximum length of a sequence due to a single SNP with $k$-mer size of 31 as 31-mers covering the SNP can extend to a maximum of 30 bases on either side of the SNP). In addition to SNPs we find associations to sites with indels and structural variations. Furthermore, we find sequences that map to multiple regions in the genome indicating copy number variations or sequence variation in repeated regions where genotype calling is known to be difficult. Although the majority of the sequences map outside of genes, we find variants in genes including in coding regions (*Figure 3(b)*).

We performed similar analysis on sequencing reads available from 87 BEB and 110 TSI individuals from the 1000 genomes project and obtained 529,287 and 462,122 sequences associated with BEB and TSI samples respectively, much fewer than the YRI-TSI comparison. *Appendix 1—figure 7* shows breakdown of probable variant types corresponding to the sequences found associated with BEB and TSI samples.

Histograms of lengths of sequences obtained by merging overlapping $k$-mers show (*Appendix 1—figure 8*, *Appendix 1—figure 9*) peaks at 61 bp which is the maximum length corresponding to a single SNP for $k$-mer size of 31. We also see drops off after 98 bp in all cases providing evidence for multinucleotide mutations (MNMs) (*Harris and Nielsen, 2014*) since this is the maximum sequence length we can get when $k$-mers of size 31 are assembled with minimum overlap of 24.

## HAWK reveals sequences not in the human reference genome

As HAWK is an alignment free method for mapping associations, it is able to find associations in regions that are not in the human reference genome, hg38. The analysis resulted in 94,795 and 66,051 sequences of lengths up to 2,666 bp and 12,467 bp associated with YRI and TSI samples respectively that did not map to the human reference genome. Similarly BEB-TSI comparison yielded 19,584 and 18,508 sequences with maximum lengths of 1761 bp and 2149 bp associated with BEB and TSI respectively.

**Table 3.** Variants in genes linked to cardiovascular diseases.

Variants in genes linked to cardiovascular diseases found to be significantly more common in BEB samples compared to TSI samples. The (%) values denote fraction of individuals in the sample with the allele present. The p-values and % values are averaged over $k$-mers constituting the associated sequences.

| Gene | SN id | Variant type | Allele | p-value | %BEB | %TSI |
|---|---|---|---|---|---|---|
| APOB | rs2302515 | Missense | C | $1.30 \times 10^{-12}$ | 29.29% | 8.37% |
| APOB | rs676210 | Missense | A | $7.73 \times 10^{-25}$ | 72.93% | 33.08% |
| APOB | rs1042034 | Missense | C | $2.28 \times 10^{-23}$ | 68.67% | 31.91% |
| CYP11B2 | rs4545 | Missense | T | $1.31 \times 10^{-28}$ | 31.33% | 0.91% |
| CYP11B1 | rs4534 | Missense | T | $9.36 \times 10^{-36}$ | 33.00% | 0.91% |
| WNK4 | rs2290041 | Missense | T | $1.53 \times 10^{-14}$ | 13.24% | 0.47% |
| WNK4 | rs55781437 | Missense | T | $1.30 \times 10^{-12}$ | 15.21% | 0.91% |
| SLC12A3 | rs2289113 | Missense | T | $7.40 \times 10^{-13}$ | 8.14% | 0.00% |
| SCNN1A | rs10849447 | Missense | C | $8.67 \times 10^{-12}$ | 62.88% | 39.92% |
| ABO | - | 4 bp (CTGT) deletion | - | $1.17 \times 10^{-13}$ | 29.15% | 10.55% |
| ABO | rs8176741 | Missense | A | $2.06 \times 10^{-16}$ | 27.70% | 8.45% |
| SH2B3 | rs3184504 | Missense | C | $8.22 \times 10^{-23}$ | 92.88% | 63.87% |
| RAI1 | rs3803763 | Missense | C | $1.32 \times 10^{-12}$ | 75.86% | 51.17% |
| RAI1 | rs11649804 | Missense | A | $1.95 \times 10^{-19}$ | 81.57% | 52.79% |

DOI: https://doi.org/10.7554/eLife.32920.007

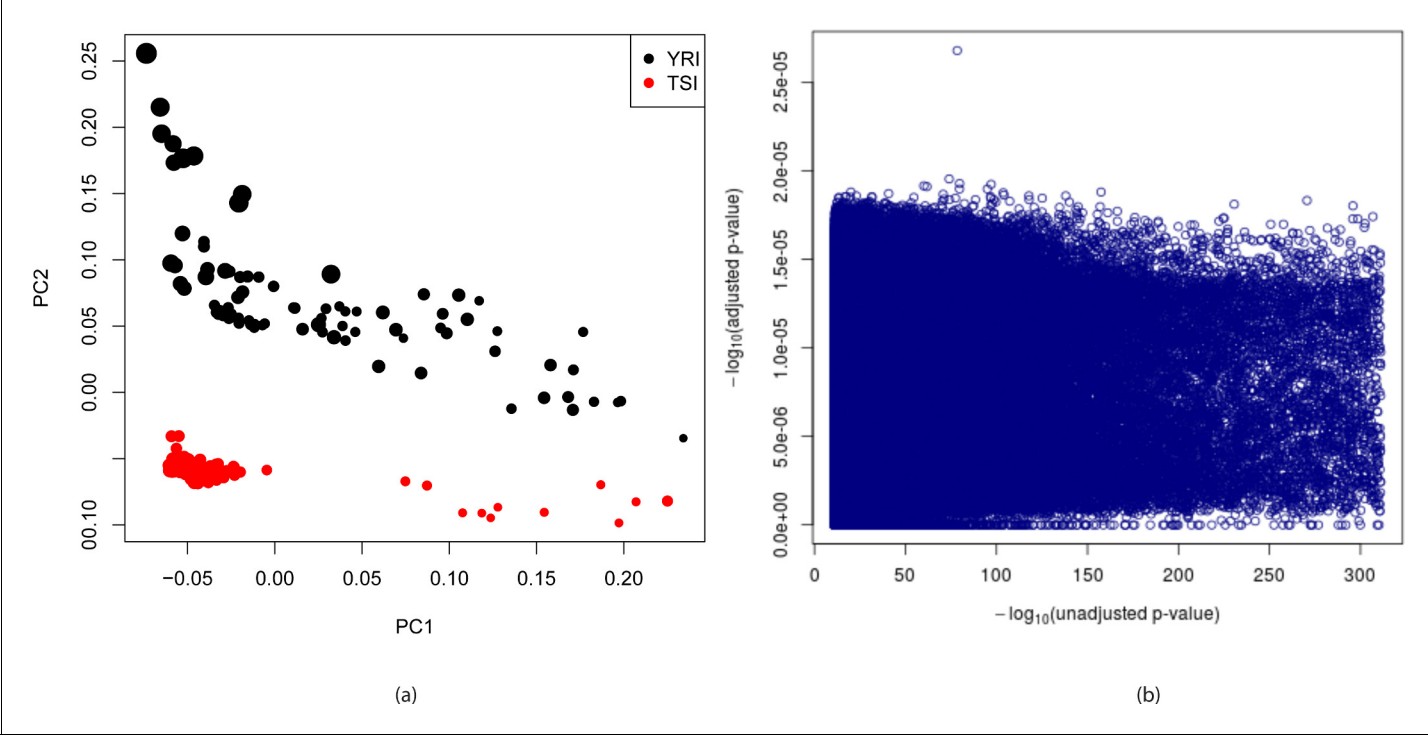

**Figure 4.** Detection and correction for population stratification in YRI-TSI dataset. (**a**) Plots of first two principal components for YRI and TSI individuals from the 1000 genomes project. The PCA was run on a binary matrix indicating presence or absence of 3,483,820 randomly chosen $k$-mers present in between 1% and 99% of the samples. The colors indicate population and sizes of circles are proportional to sequencing depth. (**b**) $-log_{10}$(adjusted p-values) are plotted against $-log_{10}$(unadjusted p-values) where adjusted p-values are calculated by fitting logistic regression models to predict population identity from $k$-mer counts adjusting for population stratification, total number of $k$-mers per sample and gender of individuals whereas unadjusted p-values are the p-values obtained using likelihood ratio test of $k$-mer counts assuming Poisson distributions.

DOI: https://doi.org/10.7554/eLife.32920.008

We found that few of the sequences enriched for in TSI samples, with lengths up to 12kbp and 2kbp in comparisons with YRI and BEB respectively, mapped to the Epstein–Barr virus (EBV) genome, strain B95-8 [GenBank: V01555.2]. EBV strain B95-8 was used to transform B cells into lymphoblastoid cell lines (LCLs) in the 1000 Genomes Project and was shown to be a contaminant in the data (*Santpere et al., 2014*).

*Table 2* summarizes the sequences that could not be mapped to either the human reference genome or the Epstein-Barr virus genome using Bowtie2. Although an exhaustive analysis of all remaining sequences using BLAST is difficult, we find sequences associated with YRI that do not map to the human reference genome (hg38) with high score but upon running BLAST aligned to other sequences from human (for example to [GenBank: AC205876.2] and some other sequences reported (*Kidd et al., 2010*). We also find sequences with no significant BLAST hits to human genomic sequences, some of which have hits to closely related species. Similarly, we find sequences associated with TSI aligning to human sequences such as [GenBank: AC217954.1] not in the reference. Although there are much fewer long sequences obtained in the BEB-TSI comparison, we find sequences longer than 1kbp associated with each population with no BLAST hit.

## Differential prevalence of variants in genes linked to CVDs in BEB-TSI comparison

We noted that cardiovascular diseases (CVD) are a leading cause of mortality in Bangladesh and age standardized death rates from CVDs in Bangladesh is higher compared to Italy (see *World Health Organization, 2011*). Moreover, South Asians have high rates of acute myocardial infarction (MI) or heart failure at younger ages compared to other populations and (*Gupta et al., 2006*; *Joshi et al., 2007*) revealed that in several countries migrants from South Asia have higher death rates from

coronary heart disease (CHD) at younger ages compared with the local population and according to the INTERHEART Study, the mean age of MI among the poeple from Bangladesh is considerably lower than non-South Asians and the lowest among South Asians (*Yusuf et al., 2004*; *Saquib et al., 2012*). This motivated us to explore probable underlying genetic causes.

The sequences of significant association with the BEB sample were aligned to RefSeq mRNAs and the ones mapping to genes linked to CVDs (*Kathiresan and Srivastava, 2012*) were analyzed. It is worth noting that the sites were obtained through a comparison of BEB and TSI samples and CVD status of the individuals were unknown. We explored whether any of the sites found due to population difference could potentially contribute towards increased mortality from CVDs in BEB. The sites listed are included as they are in genes known to be linked to CVDs but they are not highly ranked among all sites of difference between BEB and TSI.

*Table 3* shows non-synonymous variants in such genes that are significantly more common in the BEB sample compared to the TSI sample. It is worth mentioning that the 'C' allele at the SNP site, rs1042034 in the gene *Apolipoprotein B (ApoB)* has been associated with increased levels of HDL cholesterol and decreased levels of triglycerides (*Teslovich et al., 2010*) in individuals of European descent but individuals with the 'CC' genotype have been reported to have higher risk of CVDs in an analysis of the data from the Framingham Heart Study (*Kulminski et al., 2013*). Distribution of rs1042034 alleles in various populations is shown in *Appendix 1—figure 10* generated using the geography of genetic variants browser (*Marcus and Novembre, 2017*). The SNP rs676210 has also been associated with a number of traits (*Mäkelä et al., 2013*; *Chasman et al., 2009*). Both alleles of higher prevalence in BEB at those sites have been found to be common in familial hypercholesterolemia patients in Taiwan (*Chiou and Charng, 2012*). On the other hand, prevalence of the risk allele, 'T' at rs3184504 in the gene *SH2B3* is higher in TSI samples compared to BEB samples.

We also observe a number of sites in the gene *Titin (TTN)* of differential allele frequencies in BEB and TSI samples (*Appendix 1—table 1*). However, *TTN* codes for the largest known protein and although truncating mutations in *TTN* are known to cause dilated cardiomyopathy [(*Herman et al., 2012*; *van Spaendonck-Zwarts et al., 2014*; *Roberts et al., 2015*)], no such effect of other kinds of mutations are known.

## Detection and correction for confounding factors

To check whether population structure can be detected from $k$-mer data we randomly sampled approximately one thousandth of $k$-mers that appear in between 1% and 99% of the YRI and TSI datasets in the 1000 genomes project yielding 3,483,820 distinct $k$-mers and ran principal components analysis (PCA) on them. *Figure 4(a)* shows plots of the first two principal components of the individuals. Although the clusters generated are not as clearly separated as in the case with PCA run on variant calls obtained from the 1000 genomes project, shown in *Appendix 1—figure 11*, possibly due to varying sequencing coverage and batch effects in the $k$-mer counts, we observe that the first two principal components together completely separates the two populations. The first principal component correlates with sequencing depth indicated by size of the circles in the figure with the second principal component primarily separating the populations.

We then fit logistic regression models to predict population identities using first two principal components, total number of $k$-mers in each sample and gender of individuals along with $k$-mer counts and obtain ANOVA p-values of the $k$-mer counts using $\chi^2$-tests. Total number of $k$-mers was included in the model as sequencing biases such as the GC content bias are known which may lead to false positives if the cases and controls are sequenced at different sequencing depths while the genders of individuals are included to prevent false positives for $k$-mers from X and Y chromosomes in case of sex imbalance in cases and controls. Negative ten based logarithms of unadjusted p-values were calculated using a Poisson distribution based likelihood ratio test and are shown against those of adjusted p-values for 2,113,327 randomly chosen $k$-mers with statistically significant unadjusted p-values in *Figure 4(b)*. It shows that all of the $-log_{10}$(adjusted p-values) are close to zero which is expected since the first two principal components together completely separate the populations. QQ plot of adjusted p-values of 56,119 randomly chosen $k$-mers is shown in *Appendix 1—figure 11 (c)*. *Appendix 1—figure 11(d)* shows that only the first two principal components provide substantial adjustment in p-values for this dataset indicating sequencing depth and sex are not significant confounders while QQ plot for unadjusted p-values obtained from logistic regression is shown in *Appendix 1—figure 11(b)*.

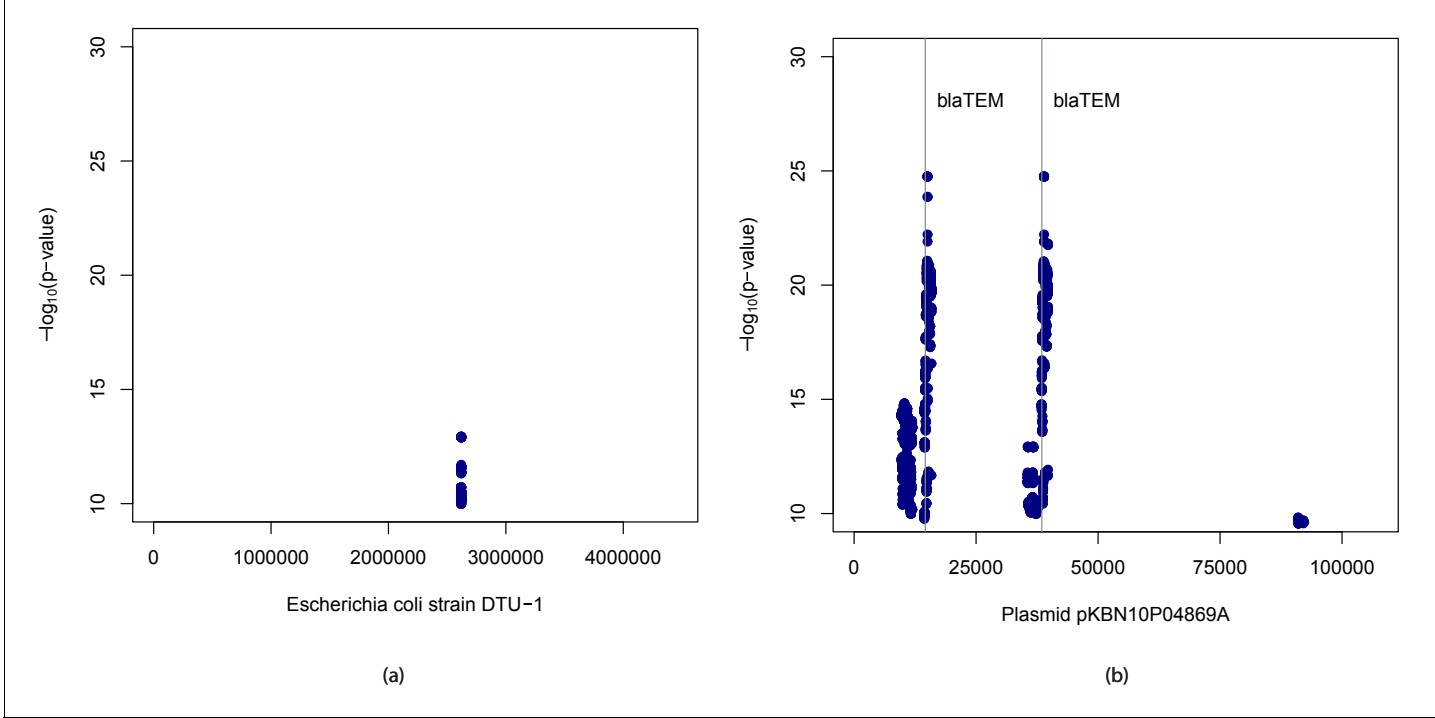

**Figure 5.** Manhattan plots for association mapping of ampicillin resistance in *E. coli* using *k*-mers. Manhattan plots showing $-log_{10}$(adjusted p-values) of *k*-mers found significantly associated with ampicillin resistance and their start positions in (a) *Escherichia coli* strain DTU-1 genome and (b) plasmid pKBN10P04869A sequence. The vertical lines denote start positions of *β*-lactamase TEM-1 gene, the presence of which is known to confer resistance to ampicillin.

DOI: https://doi.org/10.7554/eLife.32920.009

The following source data is available for figure 5:

**Source data 1.** Source data for *Figure 5a*.
DOI: https://doi.org/10.7554/eLife.32920.010
**Source data 2.** Source data for *Figure 5b*.
DOI: https://doi.org/10.7554/eLife.32920.011

We also performed simulation experiments to test whether associations can be detected after correcting for confounding factors. We set a *k*-mer as present in a YRI individual with probability $p$ and in a TSI individual with probability $1 - p$ and counts were simulated using total numbers of *k*-mers in the samples assuming Poisson distribution. The individuals with the *k*-mer were randomly assigned to cases according to penetrance values and the rest were assigned to controls. A p-value was then computed as above correcting for population stratification and other confounders and tested for significance. The process was repeated 1000 times for a particular $p$ and penetrance and repeated for other values. The fraction of runs association was detected are shown in *Appendix 1—figure 12*. We observe that with logistic regression based test that has less power compared to Poisson distribution based likelihood ratio test, associations can be detected with small sample sizes such as present ones under various conditions. For example, if 80% of case individuals are YRI, associations can be detected in all trials if penetrance is 100% and about 80% trials when it is 80%.

## Association mapping ampicillin resistance in *E. coli*

Finally we applied HAWK to map association to ampicillin resistance in *E. coli* using a dataset described in (*Earle et al., 2016*). It contains short reads from 241 strains of *E. coli*, 189 of which were resistant to ampicillin and the remaining 52 were sensitive. We ran HAWK on 176,284,643 *k*-mers obtained from the whole genome sequencing reads, first computing p-values using likelihood ratio test assuming Poisson distribution and then adjusting p-values using first ten principal components and total number of *k*-mers per sample for the top 200,000 *k*-mers associated with cases and

controls. 5047 of the $k$-mers associated with cases passed Bonferroni correction while none of the ones associated with controls did.

The $k$-mers passing Bonferroni correction were assembled using ABySS resulting in 16 sequences associated with cases. Upon running BLAST on these sequences we found hits to *Escherichia coli* strain DTU-1 genome [GenBank: CP026612.1], *Escherichia coli* strain KBN10P04869 plasmid pKBN10P04869A sequence [GenBank: CP026474.1] as well as other sequences. We then mapped the $k$-mers found significant using these two sequences as the references to obtain their locations within them. Manhattan plots of the positions thus obtained and $-log_{10}$(adjusted p-values) of the corresponding $k$-mers are shown in *Figure 5*. We found that the strongest associations are within the *β-lactamase TEM-1 (blaTEM-1)* gene which is known to confer resistance to ampicillin (also detected by [*Earle et al., 2016*]) and just upstream of that. We also noted some other hits within the *E. coli* chromosome.

QQ plot of the p-values obtained is shown in *Appendix 1—figure 13(a)*. It may be noted that for large number of $k$-mers, that are part of the *β-lactamase TEM-1 (blaTEM-1)* gene or are in linkage disequilibrium with it, the null is not true which results in deviations from the diagonal line. However, majority of the points are close to the origin which can be seen from the cumulative distribution of p-values in *Appendix 1—figure 6(b)*.

(*Earle et al., 2016*) performed an association study of ampicillin resistance using multiple approaches - SNP calling and imputations, gene presence or absence through whole genome assembly and gene finding as well as a $k$-mer based method. Their gene presence or absence approach yielded *β-lactamase TEM-1 (blaTEM-1)* as the top hit while the best $k$-mer within the causal gene found by them was of rank 6. However, neither of the approaches are likely to scale to large genomes. We also followed a more conventional approach where first the reads were mapped using Bowtie 2 (*Langmead and Salzberg, 2012*) to the reference strain CFT073 [GenBank: AE014075.1], the same reference used by (*Earle et al., 2016*). We also included plasmid pKBN10P04869A sequence [GenBank: CP026474.1] in the reference as it includes the *β-lactamase TEM-1 (blaTEM-1)* gene. Freebayes (*Garrison and Marth, 2012*) was then used to simultaneously call variants in all the strains. We finally tested each variant for association to ampicillin resistance using Eigenstrat (*Price et al., 2006*) using first ten principal components to correct for confounders. This approach resulted in no hits with genome wide significance and Manhattan plots of the variants and their $-log_{10}$(p-values) are shown in *Appendix 1—figure 14*.

## Discussion

In this paper, we presented an alignment free method for association mapping from whole genome sequencing reads. It is based on finding $k$-mers that appear significantly more times in one set of samples compared to the other and then locally assembling those $k$-mers. Since this method does not require a reference genome, it is applicable to association studies of organisms with no or incomplete reference genome. Even for human our method is advantageous as it can map associations in regions not in the reference or where variant calling is difficult.

We tested our method by applying it to data from the 1000 genomes project and comparing the results with the results obtained using the genotypes called by the project as well as using simulated data. We observe that more than 80% of the sites found using genotype calls are covered by some sequence obtained by our method while also mapping associations to regions not in the reference and in repetitive areas. Moreover, simulations suggest tests based on $k$-mer counts have more power than those based on presence and absence of alleles.

Breakdown analysis of the sequences found in pairwise comparison of YRI, TSI and BEB, TSI samples reveals that this approach allows mapping associations to SNPs, indels, structural and copy number variations through the same pipeline. In addition we find 2–4% of associated sequences are not present in the human reference genome some of which are longer than 1kbp. The YRI, TSI comparison yields almost 60kbp sequence associated with the YRI samples in sequences of length greater than 1kbp alone. This indicates populations around the world have regions in the genome not present in the reference emphasizing the importance of a reference free approach.

We explored variants in genes linked to cardiovascular diseases in the BEB, TSI comparison as South Asians are known to have a higher rate of mortality from heart diseases compared to many other populations. We find a number of non-synonymous mutations in those genes are more

common in the BEB samples in comparison to the TSI ones underscoring the importance of association studies in diverse populations. The SNP rs1042034 in the gene *Apolipoprotein B (ApoB)* merits particular mention as the CC genotype at that site has been associated with higher risk of CVDs.

We also outlined an approach to uncover population stratification, a known confounder in association studies, from $k$-mer data and correct for it and other confounders in our $k$-mer based association mapping pipeline. Application of the pipeline to map associations to ampicillin resistance in *E. coli* lead to hits to a gene, the presence of which is known to provide the resistance.

The results on simulated data, real data from the 1000 genomes project and *E. coli* datasets provide a proof of principle of this approach and motivate extension of this method to quantitative phenotypes and modeling of randomness of counts in population stratification detection and correction of confounder steps and then application to association studies of disease phenotypes in humans and other organisms.

## Acknowledgements

We thank Faraz Tavakoli, Harold Pimentel, Brielin Brown and Nicolas Bray for helpful conversations in the development of the method for association mapping from sequencing reads using $k$-mers. AR, IH, MBE and LP were funded in part by NIH R21 HG006583. AR was funded in part by Fulbright Science and Technology Fellowship 15093630.

## Additional information

### Funding

| Funder | Grant reference number | Author |
| --- | --- | --- |
| National Institutes of Health | NIH R21 HG006583 | Atif Rahman<br>Ingileif Hallgrímsdóttir<br>Michael Eisen<br>Lior Pachter |
| Fulbright Science and Technology Fellowship | 15093630 | Atif Rahman |

The funders had no role in study design, data collection and interpretation, or the decision to submit the work for publication.

### Author contributions

Atif Rahman, Conceptualization, Resources, Data curation, Software, Formal analysis, Validation, Investigation, Visualization, Methodology, Writing—original draft, Writing—review and editing; Ingileif Hallgrímsdóttir, Validation, Methodology, Writing—review and editing; Michael Eisen, Conceptualization, Supervision, Funding acquisition, Methodology, Project administration, Writing—review and editing; Lior Pachter, Conceptualization, Formal analysis, Supervision, Funding acquisition, Validation, Methodology, Project administration, Writing—review and editing

### Author ORCIDs

Atif Rahman http://orcid.org/0000-0003-1805-3971
Michael Eisen http://orcid.org/0000-0002-7528-738X

### Decision letter and Author response

Decision letter https://doi.org/10.7554/eLife.32920.032
Author response https://doi.org/10.7554/eLife.32920.033

## Additional files

### Supplementary files

• Transparent reporting form
DOI: https://doi.org/10.7554/eLife.32920.012

## Data availability

All data generated or analysed during this study are included in the manuscript and supporting files. Source data files have been provided for Figures 5.

The following previously published dataset was used:

| Author(s) | Year | Dataset title | Dataset URL | Database, license, and accessibility information |
|---|---|---|---|---|
| 1000 Genomes Project Consortium | 2012 | 1000 Genomes : A Deep Catalog of Human Genetic Variation | http://www.international-genome.org/data | Publicly available at the International Genome Sample Resource |

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

## Appendix 1

DOI: https://doi.org/10.7554/eLife.32920.013

### Finding differential $k$-mers in association studies

We consider the case where we have $s_1$ and $s_2$ samples from two populations. We observe a specific $k$-mer $k_{1,1}, \ldots, k_{1,s_1}$ and $k_{2,1}, \ldots, k_{2,s_2}$ times in the samples from two populations and total $k$-mer counts in the samples are given by $n_{1,1}, \ldots, n_{1,s_1}$ and $n_{2,1}, \ldots, n_{2,s_2}$. We assume that the $k$-mer counts are Poisson distributed with rate $\theta_1$ and $\theta_2$ in the two populations where the $\theta$'s can be interpreted as quantities proportional to the average number of times the $k$-mer appears in the two populations. The null hypothesis is $H_0{:}\theta_1 = \theta_2 = \theta$ and the alternate hypothesis is $H_1{:}\theta_1 \neq \theta_2$

We test the null using likelihood ratio test for nested models. The likelihood ratio is given by

$$\Lambda = \frac{sup\{L(\theta_1, \theta_2)\}}{sup\{L(\theta, \theta)\}}$$

where

$$L(\theta_1, \theta_2) = \prod_{i=1}^{s_1} \frac{e^{-\theta_1 n_{1,i}} \left(\theta_1 n_{1,i}\right)^{k_{1,i}}}{k_{1,i}!} \prod_{i=1}^{s_2} \frac{e^{-\theta_2 n_{2,i}} \left(\theta_2 n_{2,i}\right)^{k_{2,i}}}{k_{2,i}!}$$

and

$$L(\theta) = \prod_{i=1}^{s_1} \frac{e^{-\theta n_{1,i}} \left(\theta n_{1,i}\right)^{k_{1,i}}}{k_{1,i}!} \prod_{i=1}^{s_2} \frac{e^{-\theta n_{2,i}} \left(\theta n_{2,i}\right)^{k_{2,i}}}{k_{2,i}!}.$$

$L(\theta_1, \theta_2)$ is maximized at $\hat{\theta}_1 = \frac{\sum_{i=1}^{s_1} k_{1,i}}{\sum_{i=1}^{s_1} n_{1,i}}$, $\hat{\theta}_2 = \frac{\sum_{i=1}^{s_2} k_{2,i}}{\sum_{i=1}^{s_2} n_{2,i}}$ and $L(\theta)$ is maximized at $\hat{\theta} = \frac{\sum_{i=1}^{s_1} k_{1,i} + \sum_{i=1}^{s_2} k_{2,i}}{\sum_{i=1}^{s_1} n_{1,i} + \sum_{i=1}^{s_2} n_{2,i}}$. Therefore,

$$\Lambda = \frac{L\left(\hat{\theta}_1, \hat{\theta}_2\right)}{L\left(\hat{\theta}, \hat{\theta}\right)}.$$

Since the null model is a special case of the alternate model, $2ln\Lambda$ is approximately chi-squared distributed with one degree of freedom.(**Wilks, 1938**; **Huelsenbeck and Crandall, 1997**; **Huelsenbeck et al., 1996**)

We note that the test statistic stays the same if the likelihood values are computed by pooling together the counts in samples from two populations that is

$$L(\theta_1, \theta_2) = \frac{e^{-\theta_1 N_1} (\theta_1 N_1)^{K_1}}{K_1!} \frac{e^{-\theta_2 N_2} (\theta_2 N_2)^{K_2}}{K_2!}$$

and

$$L(\theta) = \frac{e^{-\theta N_1} (\theta N_1)^{K_1}}{K_1!} \frac{e^{-\theta N_2} (\theta N_2)^{K_2}}{K_2!}$$

where $\sum_{i=1}^{s_1} k_{1,i} = K_1$, $\sum_{i=1}^{s_2} k_{2,i} = K_2$, $\sum_{i=1}^{s_1} n_{1,i} = N_1$, and $\sum_{i=1}^{s_2} n_{2,i} = N_2$.

For each $k$-mer in the data, we compute the statistic as described above and obtain a P-value using $\chi_1^2$ distribution. The P-values are then corrected for multiple testing using Bonferroni correction.

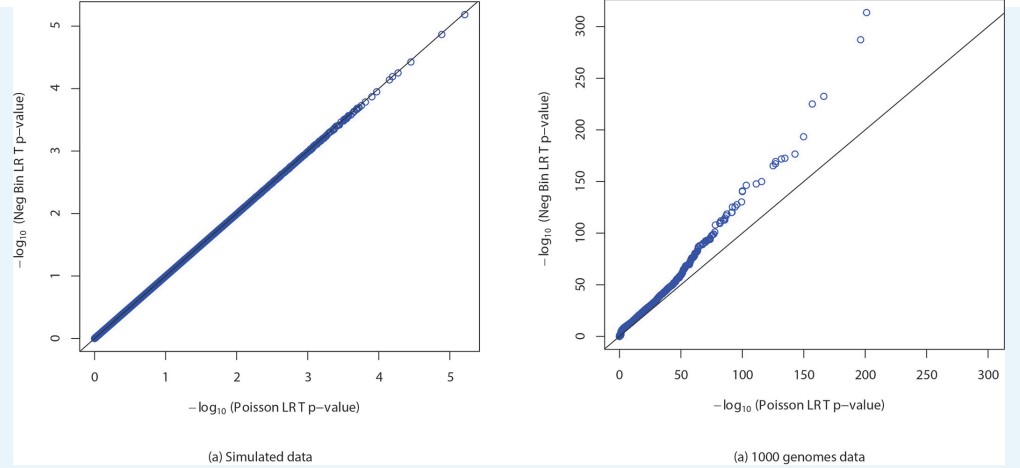

(a) Simulated data          (a) 1000 genomes data

**Appendix 1—figure 1.** QQ plots of p-values. QQ plots of p-values calculated using likelihood ratio test with Poisson and negative binomial distributions for (a) simulated data and (b) real data from the 1000 genomes project. The simulation was performed by sampling 200 values from a negative binomial distribution with number of failures fixed at 1 million and a uniformly chosen success probability between 0 and 0.01. Half of the samples were assigned to cases and others were assigned to controls. Then p-values were computed using likelihood ratio tests with Poisson and negative binomial distributions. The process was repeated $100, 000$ times and QQ plot was generated. Similarly p-values were computed for counts of $56, 119$ randomly chosen $k$-mers from YRI and TSI populations from the 1000 genomes project.

DOI: https://doi.org/10.7554/eLife.32920.014

## Power Calculation

To evaluate the theoretical power of the test, we fixed the number of times a $k$-mer is present in control samples at 0, obtained the minimum $k$-mer count in case samples required to obtain a p-value less than significant threshold after Bonferroni correction and calculated the probability of observing at least that count in case samples for varying total $k$-mer coverage of cases (number of case samples $\times k$-mer coverage per sample). The results are plotted in *Appendix 1—figure 2* for different total number of tests.

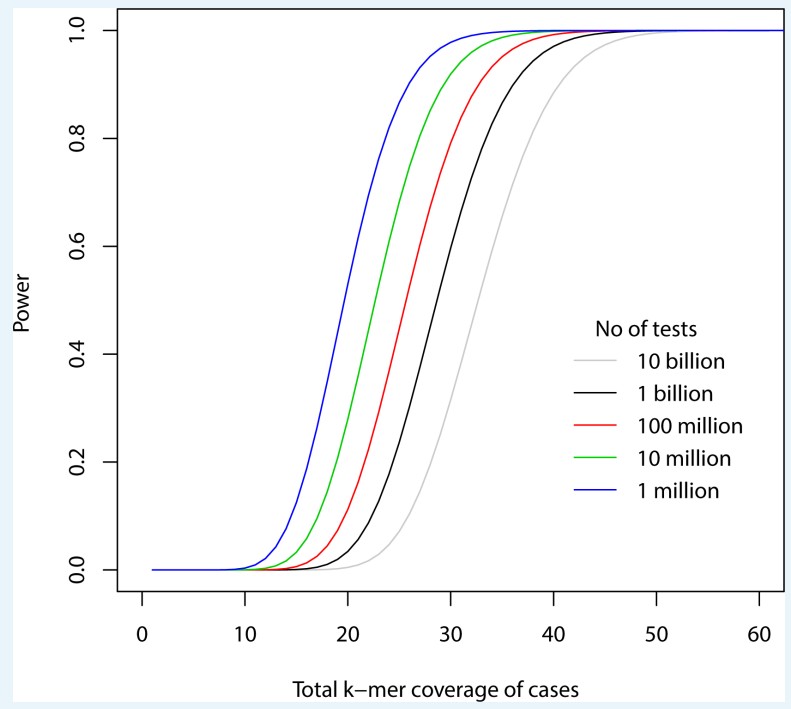

**Appendix 1—figure 2.** Power for different $k$-mer coverages. The figure shows power to detect a $k$-mer present in all case samples and no control sample against total $k$-mer coverage of cases using Bonferroni correction for different number of total tests for p-value=0.05.

DOI: https://doi.org/10.7554/eLife.32920.015

We have also performed a simulation similar to the one in **Lees et al. (2016)**. Ratio of cases to controls was set at 0.5. Then for a fixed minor allele frequency (MAF) and different odds ratios (OR) and total number of samples, the number of case samples with the variant was determined by solving a quadratic equation and the probabilities of samples being case or control with and without the variant were calculated. Cases and controls were then sampled 100 times using those probabilities. The fraction of times the p-value obtained was below the significance level, after Bonferroni correction for 1 million tests, are then plotted against the number of samples in **Appendix 1—figure 3**.

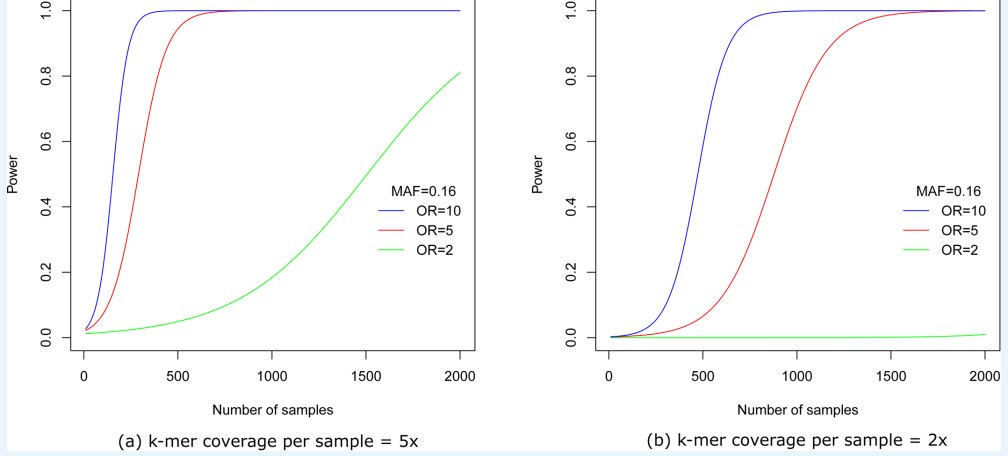

(a) k-mer coverage per sample = 5x   (b) k-mer coverage per sample = 2x

**Appendix 1—figure 3.** Power vs number of samples. Figures show power to detect a $k$-mer at MAF = 0.16 and different odds ratios for per sample $k$-mer coverage of (**a**) five and (**b**) two after Bonferroni correction for 1 million tests.

DOI: https://doi.org/10.7554/eLife.32920.016

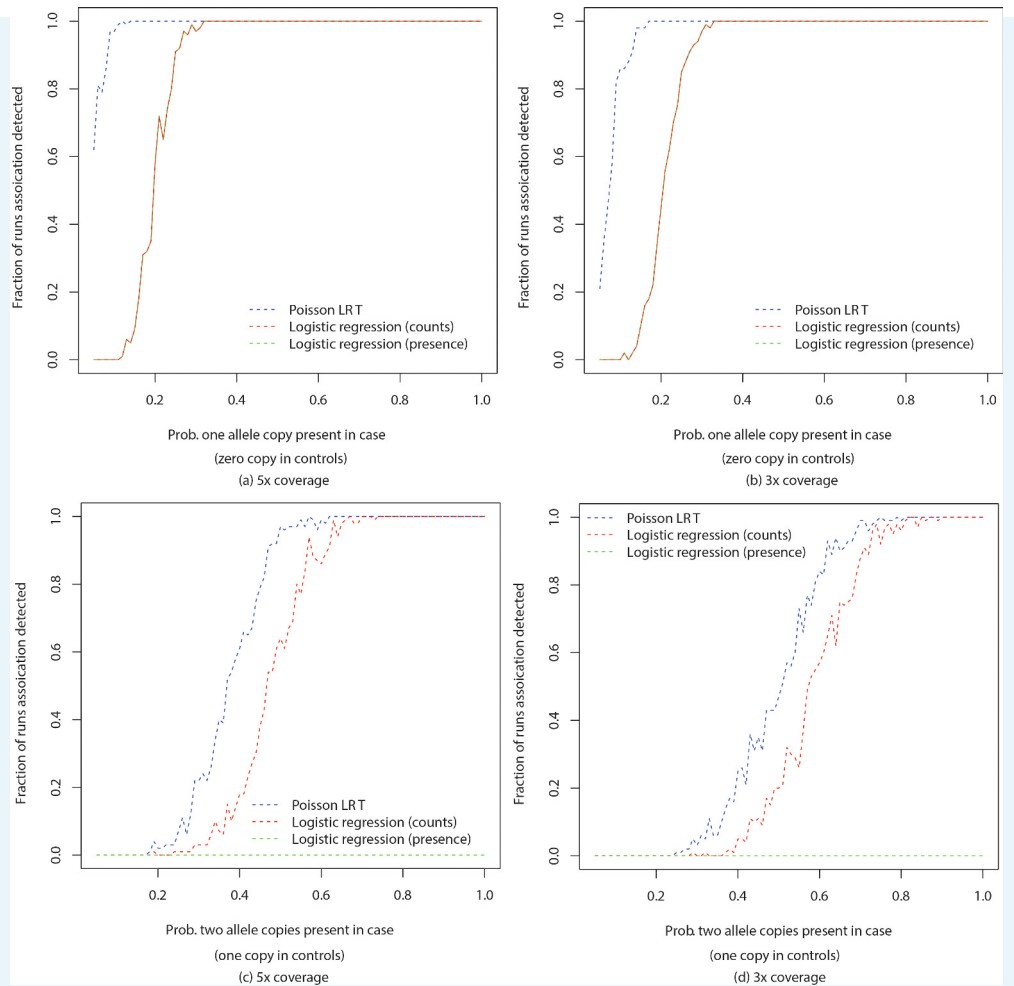

**Appendix 1—figure 4.** Comparison of powers of Poisson LRT and logistic regression based tests. Figures show power to detect a $k$-mer by Poisson based likelihood ratio test, logistic regression based tests using $k$-mer counts and presence and absence of $k$-mers with number of allele copies in controls fixed at (**a**), (**b**) 0 and (**c**), (**d**) 1. Simulation was performed by randomly choosing case individuals with specified number of copies of an allele according to varying probability. $k$-mer counts were then generated according to Poisson distribution and detection of association was checked after Bonferroni correction for 1 million tests. The process was repeated 100 times for each set of parameters. Number of cases and number of controls were fixed at 100.

DOI: https://doi.org/10.7554/eLife.32920.017

## Verification with simulated E. coli data

The simulation with *E. coli* genome (∼ 4.6 million bp) was performed by first introducing 100 single base changes, 100 indels of random lengths less than 10 bp and 100 indels of random lengths between 100 and 1000 bp at random locations. Then different number of controls and cases were generated using wgsim of SAMtools (*Li et al., 2009*) introducing more mutations with default parameters and 300000 paired end reads of length 70 (∼ 5x $k$-mer coverage) were generated with sequencing error rate of 0.01.

The sensitivity and specificity analysis was done by aligning the sequences generated by HAWK using Bowtie 2 (*Langmead and Salzberg, 2012*) and checking for overlap with mutation locations with in-house scripts.

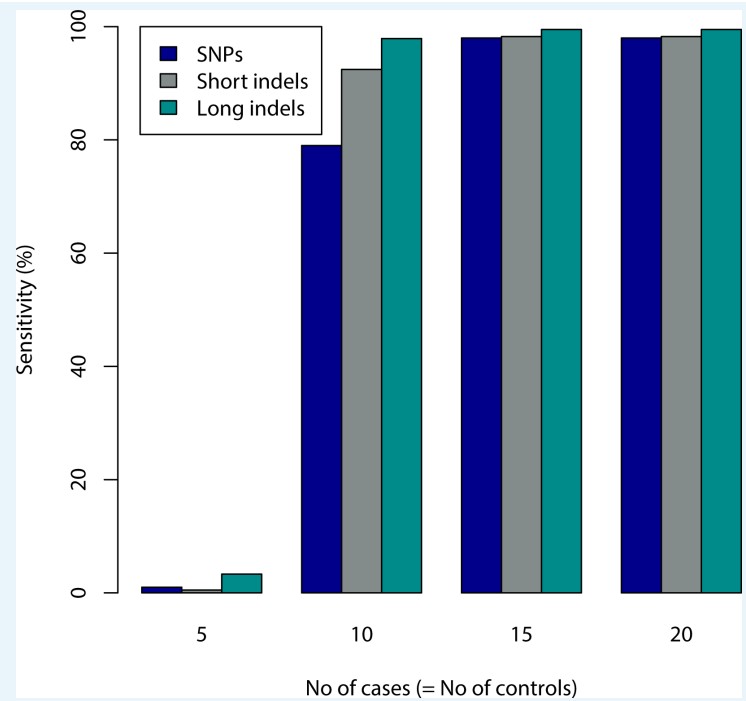

**Appendix 1—figure 5.** Sensitivity with simulated *E. coli* data. The figure shows sensitivity for varying number of case and control samples for different types of mutations. Sensitivity is defined as the percentage of differing nucleotides that are covered by a sequence. All of the sequences covered some location of mutation.

DOI: https://doi.org/10.7554/eLife.32920.018

## Testing for significant genotypes

Consider a site with two alleles. Let $n_{1,0}, n_{1,1}, n_{1,2}$ be the number of individuals with 0,1 and 2 copies of the minor allele respectively in the sample from population one and $n_{2,0}, n_{2,1}, n_{2,2}$ are the corresponding ones from population 2. Let $p_1$ and $p_2$ be the minor allele frequencies in the two populations and $N_1$ and $N_2$ be the number of samples. The null hypothesis is $H_0: p_1 = p_2 = p$ and the alternate hypothesis is $H_0: p_1 \neq p_2$

We test the null using likelihood ratio test for nested models. The likelihood ratio is given by

$$\Lambda = \frac{sup\{L(p_1, p_2)\}}{sup\{L(p, p)\}}$$

where under random mating

$$L(p_1, p_2) = \binom{N_1}{n_{1,0}, n_{1,1}, n_{1,2}} (1 - p_1)^{2n_{1,0}} (2p_1(1 - p_1))^{n_{1,1}} p_1^{2n_{1,2}}$$
$$\binom{N_2}{n_{2,0}, n_{2,1}, n_{2,2}} (1 - p_2)^{2n_{2,0}} (2p_2(1 - p_2))^{n_{2,1}} p_2^{2n_{2,2}}$$

and

$$L(p) = \binom{N_1}{n_{1,0}, n_{1,1}, n_{1,2}} (1 - p)^{2n_{1,0}} (2p(1 - p))^{n_{1,1}} p^{2n_{1,2}}$$
$$\binom{N_2}{n_{2,0}, n_{2,1}, n_{2,2}} (1 - p)^{2n_{2,0}} (2p(1 - p))^{n_{2,1}} p^{2n_{2,2}}.$$

$L(p_1, p_2)$ is maximized at $\hat{p}_1 = \frac{n_{1,1} + 2n_{1,2}}{2N_1}$, $\hat{p}_2 = \frac{n_{2,1} + 2n_{2,2}}{2N_2}$ and $L(p)$ is maximized at $\hat{p} = \frac{n_{1,1} + 2n_{1,2} + n_{2,1} + 2n_{2,2}}{2N_1 + 2N_2}$. Therefore,

$$\Lambda = \frac{L(\hat{p}_1, \hat{p}_2)}{L(\hat{p}, \hat{p})}.$$

Since the null model is a special case of the alternate model, $2ln\Lambda$ is approximately chi-squared distributed with one degree of freedom.

For each SNP site in the data, we compute the statistic as described above and obtain a p-value using $\chi_1^2$ distribution. The p-values are then corrected for multiple testing using Bonferroni correction.

## Intersection analysis using 1000 genomes data

For the intersection analysis we used data from 87 YRI individuals and 98 TSI individuals for which both sequencing reads and genotype calls were available. The HAWK pipeline was run using a $k$-mer size of 31. The assembly was using ABySS (**Simpson et al., 2009**). The sequences were aligned to the human reference genome version GRCh37 using Bowtie2. Each of the genotypes called by the 1000 genomes project was then tested for association with populations using the approach described in the previous section. The extent of intersection of the loci found using two methods was then determined using BEDtools (**Quinlan and Hall, 2010**).

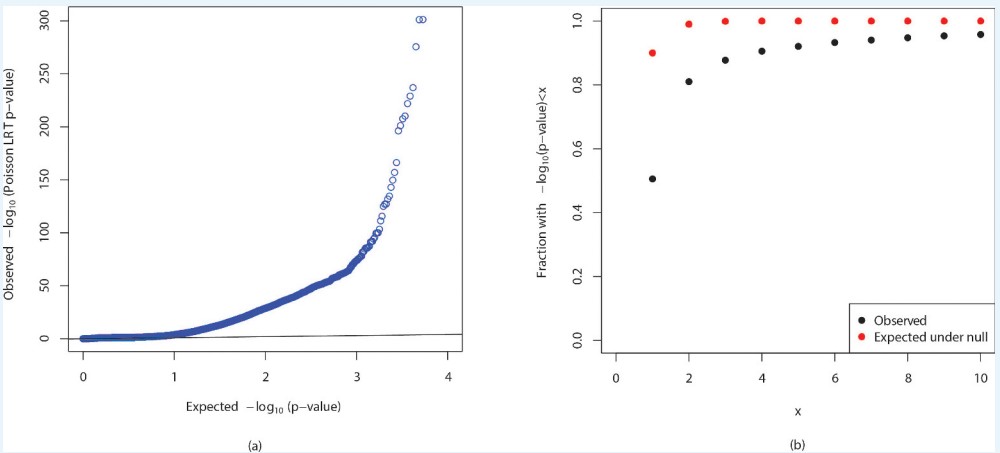

(a)                                        (b)

**Appendix 1—figure 6.** QQ plot and cumulative distributions of p-values. The figures show (a) QQ plot and (b) cumulative distributions of $-log_{10}$(p-values) for observed p-values in the comparison of YRI and TSI populations from the 1000 genomes project and p-values expected if the null is true.

DOI: https://doi.org/10.7554/eLife.32920.019

## Breakdown analysis

The types of variants corresponding to sequences found using the HAWK pipeline were estimated by mapping them to the human reference genome version hg38 using Bowtie2 and using following properties.

- **Unmapped:** The sequences that were not mapped to the reference using Bowtie2.
- **Multimapped:** The sequences with multiple mappings. These may be due to copy number variations or sequence variation in repetitive regions.
- **Indels:** The sequences that mapped to the reference with one or more indels.
- **Multiple SNPs/Structural:** The maximum length of a sequence due to a single SNP with 31-mer is 61. Sequences with length greater than 61 were assigned this label.
- **SNPs:** All other sequences.

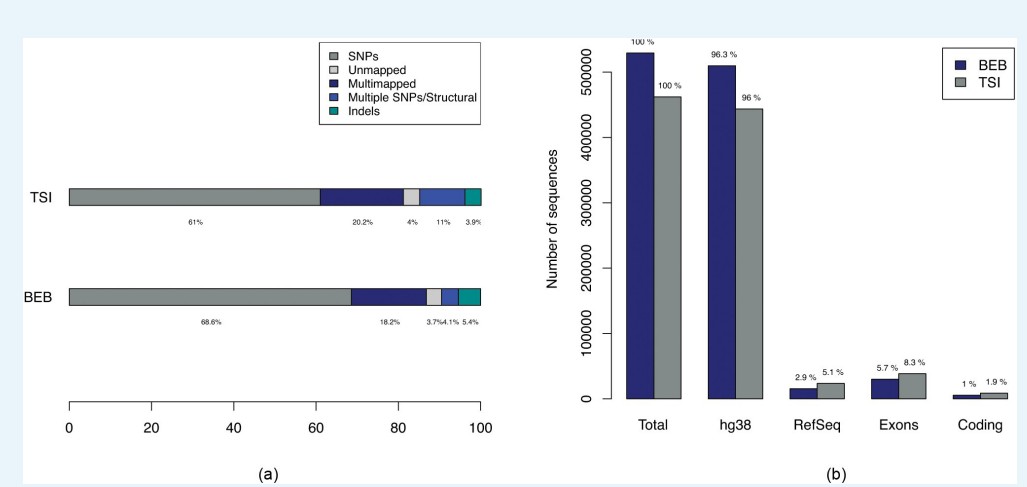

**Appendix 1—figure 7.** Breakdown of types of variations in BEB-TSI comparison. (**a**) Bar plots showing breakdown of 529,287 and 462,122 sequences associated with BEB and TSI samples respectively. The 'Multiple SNPs/Structural' entries correspond to sequences of length greater than 61, the maximum length of a sequence due to a single SNP with $k$-mer size of 31 and 'SNPs' correspond to sequences of maximum length of 61. (**b**) Numbers of sequences with alignments to hg38, RefSeq mRNAs and Ensembl exons and coding regions.

DOI: https://doi.org/10.7554/eLife.32920.020

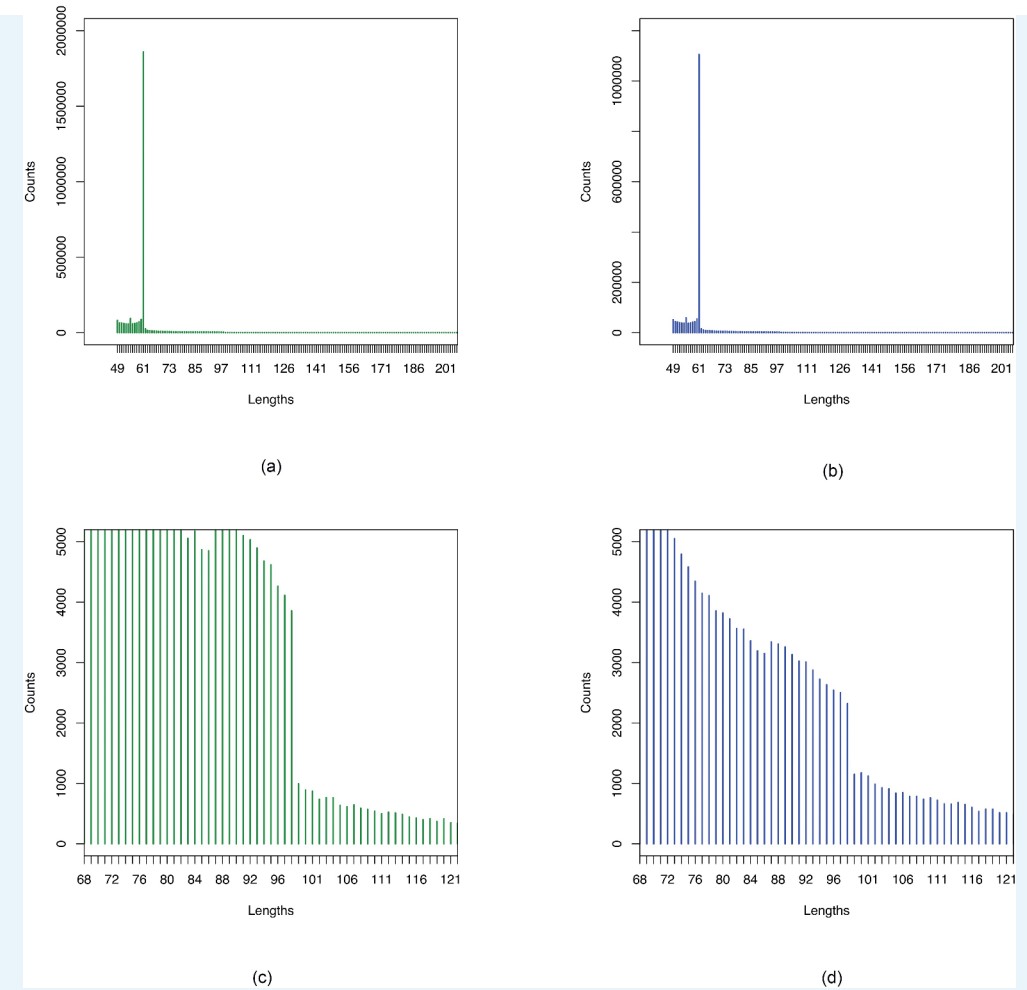

**Appendix 1—figure 8.** Histograms of sequence lengths in YRI-TSI comparison. Figures show sections of histograms of lengths of sequences associated with (**a**, **c**) YRI and (**b**, **d**) TSI in comparison of YRI and TSI samples. Figures (**a**), (**b**) show peaks at 61, the maximum length corresponding to a single SNP with $k$-mer size of 31. Figures (**c**), (**d**) show drop off after 98 which is the maximum length corresponding to two close-by SNPs as 31-mers were assembled using a minimum overlap of 24.

DOI: https://doi.org/10.7554/eLife.32920.021

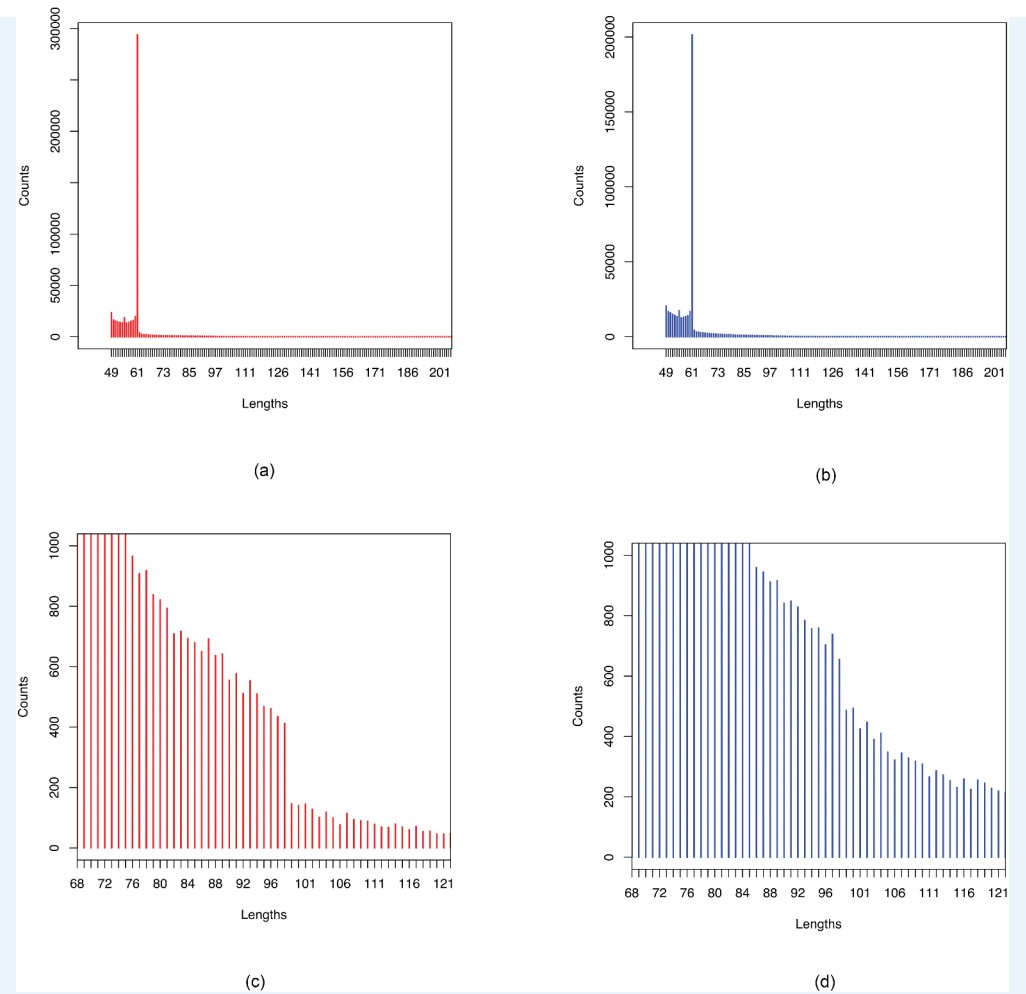

**Appendix 1—figure 9.** Histograms of sequence lengths in BEB-TSI comparison. Figures show sections of histograms of lengths of sequences associated with (**a**, **c**) BEB and (**b**, **d**) TSI in comparison of BEB and TSI samples. Figures (**a**), (**b**) show peaks at 61, the maximum length corresponding to a single SNP with $k$-mer size of 31. Figures (**c**), (**d**) show drop off after 98 which is the maximum length corresponding to two close-by SNPs as 31-mers were assembled using a minimum overlap of 24.

DOI: https://doi.org/10.7554/eLife.32920.022

## Probing for variants of interest

We searched for well known variants and other variants of potential biological interest by aligning sequences to RefSeq mRNAs using Bowtie2 and then looking up gene names from RefSeq mRNA identifiers using the UCSC Table Browser (*Karolchik et al., 2004*). Variants of interest were then further explored by running BLAT (*Kent, 2002*) on the UCSC Human Genome Browser (*Kent et al., 2002*).

chr2:21225281 C/T

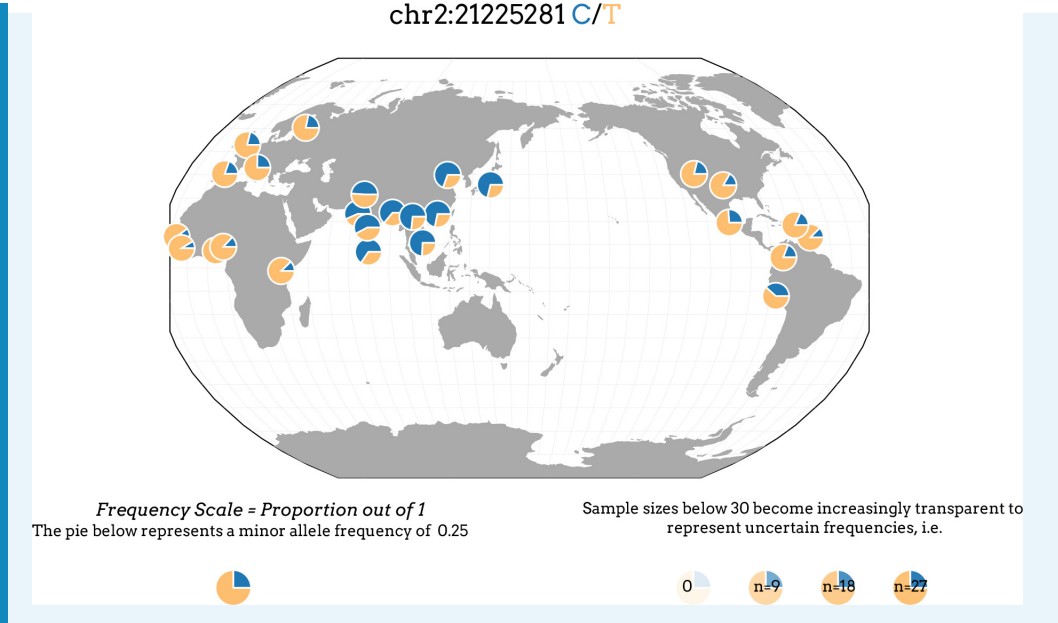

Appendix 1—figure 10. Distribution of the SNP rs1042034 alleles in 1000 genomes populations. Plot obtained from the Geography of Genetic Variants Browser showing distribution of alleles at the SNP site rs1042034 revealing the 'C' allele is more prevalent in South, Southeast and East Asian populations compared to populations from other parts of the world.

DOI: https://doi.org/10.7554/eLife.32920.023

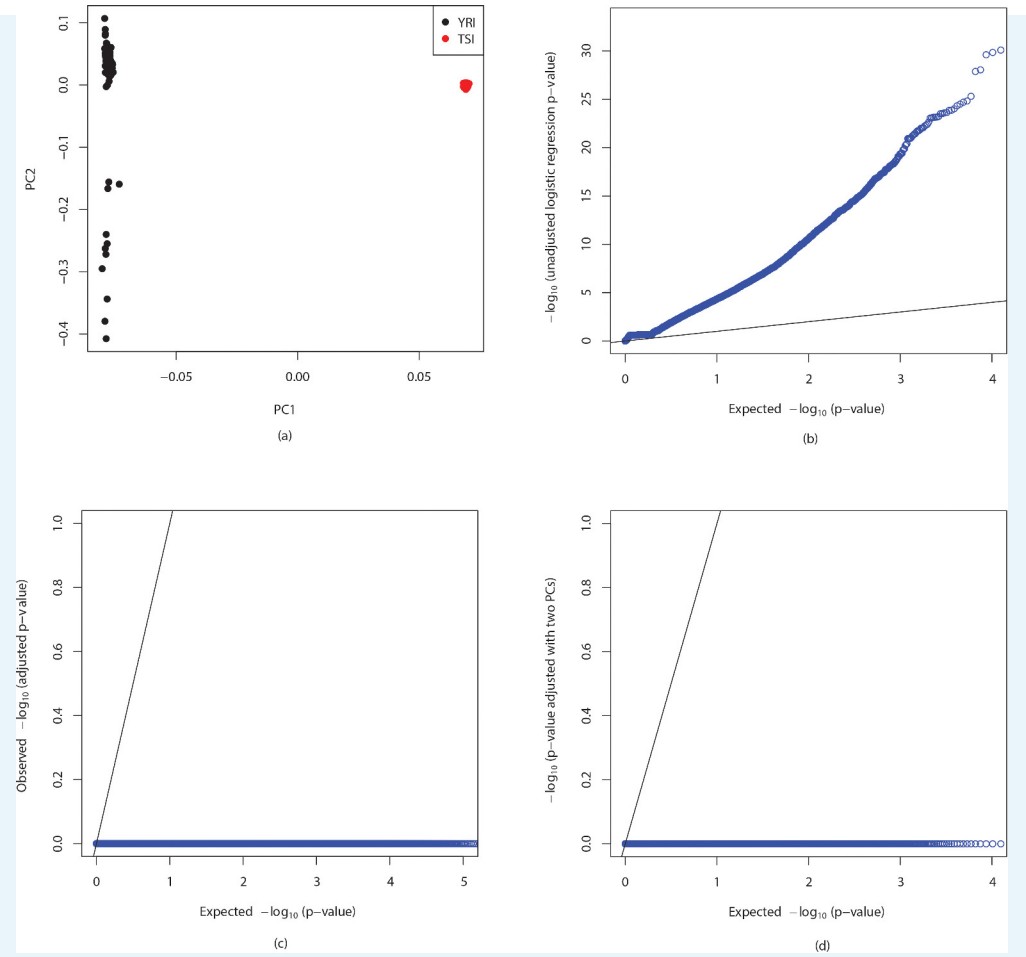

**Appendix 1—figure 11.** Detection and correction for population stratification in YRI-TSI data-set. (**a**) Plots of first two principal components for YRI and TSI individuals where the PCA was run on variants called in the 1000 genomes project. QQ plots of ANOVA $\chi^2$ p-values of $k$-mer counts computed by running logistic regression (**b**) without adjustment, (**c**) adjusted using first two principal components obtained from $k$-mer counts, total number of $k$-mers and sex of individuals, (**d**) adjusted using only first two principal components.

DOI: https://doi.org/10.7554/eLife.32920.024

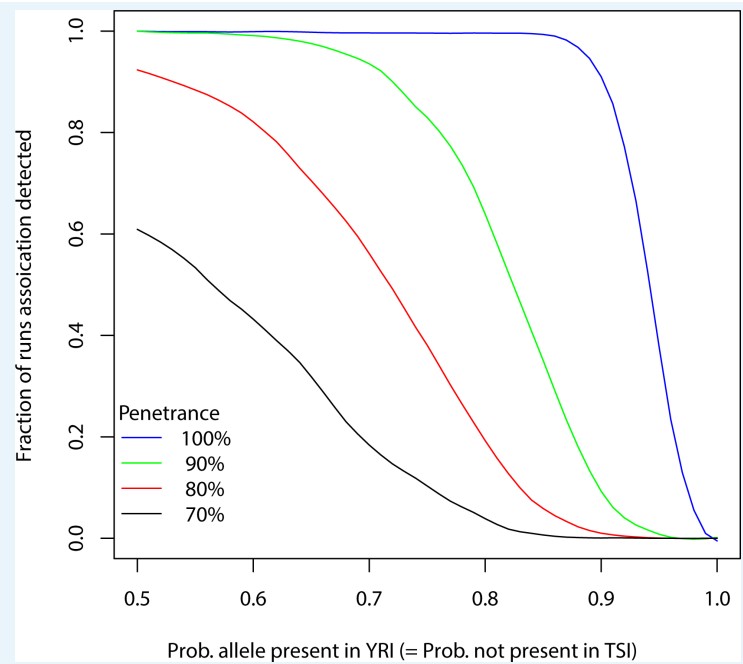

**Appendix 1—figure 12.** Detection of association after correcting for confounders. A $k$-mer (allele) is considered to be present in a YRI individual with probability $p$ and in a TSI individual with probability $1-p$ and counts are simulated using total numbers of $k$-mers in the samples assuming Poisson distribution. The individuals with the $k$-mer are randomly assigned to cases according to a penetrance value and the rest are assigned to controls. A p-value is then computed using logistic regression to predict phenotype from the counts correcting for population stratification, total number of $k$-mers per sample and gender of individuals from the 1000 genomes data. The process is repeated 1000 times for a particular $p$ and penetrance and fraction of runs where the p-value passed the Bonferroni threshold is plotted. The process is repeated for various probabilities and penetrance.

DOI: https://doi.org/10.7554/eLife.32920.025

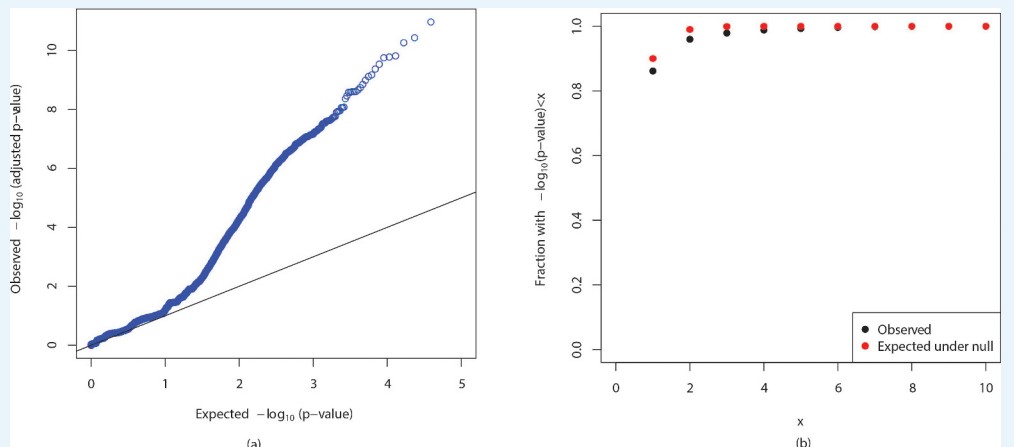

**Appendix 1—figure 13.** QQ plot and cumulative distributions of p-values. The figures show (**a**) QQ plot and (**b**) cumulative distributions of $-log_{10}$(p-values) for observed p-values in association mapping of ampicillin resistance in *E. coli* and p-values expected if the null is true.

DOI: https://doi.org/10.7554/eLife.32920.026

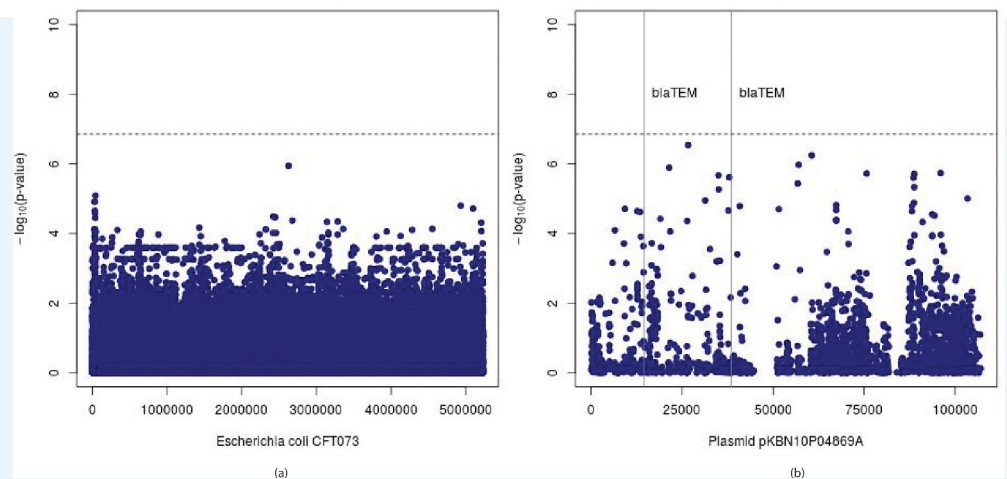

**Appendix 1—figure 14.** Manhattan plots for association mapping of ampicillin resistance in *E. coli* using conventional approach. Manhattan plots showing $-\log_{10}$(p-values) of SNPs and their positions in (**a**) *Escherichia coli* strain CFT073 genome and (**b**) plasmid pKBN10P04869A sequence. The vertical lines denote start positions of *β-lactamase TEM-1* gene, the presence of which is known to confer resistance to ampicillin. The horizontal lines denote Bonferroni threshold of 0.05/361293.
DOI: https://doi.org/10.7554/eLife.32920.027

**Appendix 1—table 1.** Variants in *Titin* of differential prevalence in BEB-TSI comparison. Variants in *Titin*, a gene linked to cardiovascular diseases, that were found to be significantly more common in BEB samples compared to TSI samples. The (%) values denote fraction of individuals in the sample with the allele present. The p-values and % values are averaged over *k*-mers constituting the associated sequences.

| Gene | SNP id | Variant type | Allele | p-value | %BEB | %TSI |
|---|---|---|---|---|---|---|
| TTN | rs9808377 | Missense | G | $1.70\times10^{-15}$ | 66.44% | 41.91% |
| TTN | rs62621236 | Missense | G | $2.33\times10^{-16}$ | 27.70% | 5.72% |
| TTN | rs2291311 | Missense | C | $1.06\times10^{-11}$ | 25.77% | 7.77% |
| TTN | rs16866425 | Missense | C | $8.19\times10^{-12}$ | 21.73% | 2.73% |
| TTN | rs4894048 | Missense | T | $2.00\times10^{-23}$ | 22.65% | 2.26% |
| TTN | rs13398235 | Intron/missense | A | $2.04\times10^{-13}$ | 41.00% | 17.40% |
| TTN | rs11888217 | Intron/missense | T | $4.18\times10^{-13}$ | 27.25% | 4.55% |
| TTN | rs10164753 | Missense | T | $3.69\times10^{-13}$ | 28.48% | 6.19% |
| TTN | rs10497520 | Missense | T | $1.66\times10^{-23}$ | 54.76% | 18.86% |
| TTN | rs2627037 | Missense | A | $6.99\times10^{-13}$ | 25.06% | 4.72% |
| TTN | rs1001238 | Missense | C | $1.66\times10^{-17}$ | 64.66% | 38.21% |
| TTN | rs3731746 | Missense | A | $1.26\times10^{-14}$ | 50.72% | 30.21% |
| TTN | rs17355446 | Intron/missense | A | $3.31\times10^{-11}$ | 15.44% | 1.11% |
| TTN | rs2042996 | Missense | A | $1.03\times10^{-17}$ | 71.41% | 35.87% |
| TTN | rs747122 | Missense | T | $1.59\times10^{-11}$ | 28.57% | 7.24% |
| TTN | rs1560221* | Synonymous | G | $1.11\times10^{-22}$ | 70.71% | 34.66% |
| TTN | rs16866406 | Missense | A | $2.17\times10^{-12}$ | 35.60% | 17.51% |
| TTN | rs4894028 | Missense | T | $2.58\times10^{-13}$ | 27.54% | 6.89% |
| TTN | - | Insertion | T | $1.09\times10^{-12}$ | 34.11% | 8.59% |

*Appendix 1—table 1 continued on next page*

*Appendix 1—table 1 continued*

| Gene | SNP id | Variant type | Allele | p-value | %BEB | %TSI |
|------|--------|-------------|--------|---------|------|------|
| TTN | rs3829747 | Missense | T | $4.72\times10^{-12}$ | 37.55% | 20.30% |
| TTN | rs2291310 | Missense | C | $2.18\times10^{-20}$ | 36.63% | 8.04% |
| TTN | rs2042995 | Intron/Missense | C | $7.83\times10^{-12}$ | 56.32% | 31.64% |
| TTN | rs3829746 | Missense | C | $5.30\times10^{-29}$ | 75.94% | 37.60% |
| TTN | rs744426 | Missense | A | $1.36\times10^{-13}$ | 37.15% | 18.92% |

DOI: https://doi.org/10.7554/eLife.32920.028

