## [Decision Letter]

Thank you for submitting your article "Association Mapping from Sequencing Reads using *k*-mers" for consideration by *eLife*. Your article has been reviewed by three peer reviewers, and the evaluation has been overseen by a Reviewing Editor and Detlef Weigel as the Senior Editor. The following individual involved in review of your submission has agreed to reveal his identity: Casey S Greene (Reviewer #3).

The reviewers have discussed the reviews with one another and the Reviewing Editor has drafted this decision to help you prepare a revised submission.

Summary:

The paper describes a method and associated software (HAWK) that performs association mapping on sequencing data without the use of read alignment and variant calling. The method tests if *k*-mers are differentially represented in the reads from cases vs. controls, and concatenates overlapping *k*-mers to find out longer sequences are associated with the trait of interest. The method is particularly well suited to detecting associations with structural variants, since it avoids the problem of SV calling, which is currently challenging with current methods.

Overall, there was much enthusiasm for the work, but there were serious concerns regarding how the authors would account for population structure. This will need to be thoroughly addressed in a revision.

Essential revisions:

1) Demonstrate that the method is robust to problems of population structure.

2) Statistical problem: to test if *k*-mers are differentially represented in cases vs. controls, the authors do a simple two-sample Poisson comparison using likelihood ratio test. This makes the implicit assumption that the read counts are Poisson distributed and the Poisson rates are constant in cases and in controls. The latter assumption is generally problematic for read count data, as biological differences can create additional variations of read counts. Please show empirical evidence that the counts are Poisson, and indicate what could go wrong if they are not.

It is known in RNA-seq data, for example, that read counts from multiple samples cannot be adequately modeled by a Poisson distribution, and a negative binomial distribution is used to model "overdispersion" (some relevant papers are edgeR (Small-sample estimation of negative binomial dispersion, with applications to SAGE data, Robinson & Smith, Biostatistics, 2008) and WASP (PMID: 26366987)).

It is not entirely clear that using negative binomial (NB) would solve the problem in this case, as the underlying Poisson rates may be discrete. For instance, suppose the association signal is driven by a deletion (CNV). Each individual either has the deletion or not. NB makes the assumption that the rate follows a gamma distribution. The authors should show the signal is not driven by overdispersion or confounders. At the minimum, they should have a QQ plot, which shows that for most *k*-mers, there is no association.

3) A related problem is the possible confounders. For example, if cases and controls have different coverage, and representation of *k*-mers somehow depends on coverage (GC content is correlated with coverage, for example), one could imagine that this may create some false associations.

4) The validation of the method is inadequate. The experiment of classifying population ancestry is a quite artificial – few people will do this kind of analysis. What is much more relevant is application of the method on real sequencing data of some complex phenotype. The authors did do it on a CVD dataset, but only superficially. They applied it only on genes known to be linked to CVD, for example, and it's unclear how the results compare with the alternative approach. To really demonstrate the benefits of HAWK, they should apply it to the full data, and evaluate the results more deeply. How do the results compare with standard GWAS? Any new associations? Can these new associations by validated? Do we learn any biology?

5) The authors state that Cortex (Iqbal et al.) is not suited to a large number of individuals (Introduction, third paragraph). Please explain why this is the case. A comparison with HAWK would help to motivate their approach.

6) Iqbal et al. say that "Current assembly methods also typically ignore pre-existing information, such as a reference sequence or known variants. Although variant discovery should not be biased by such information, neither should this information be discarded." The authors should comment on this with respect to their approach.

7) For the comparison between HAWK and traditional genotype GWAS (subsection “Verification with 1000 genomes data”), restrict the analysis to only SNPs, i.e. to variants that a normal GWAS is meant to capture. Then we can see the power difference afforded by Poisson and access to counts vs genotype calls. Also, include the use of dosages, a softer version of a hard variant calls. If the false positive rate is well-controlled here (provide quantile plots of p-values), and you are well-powered, then the much-harder-to-verify claims about finding structural variants are better supported.

8) The authors should discuss the error rate they are interested in and why, and then explain what an appropriate error measure would be. For example, it seems that Storey et al.'s *q*-values may be more appropriate (Bonferroni is overly conservative).

9) In the comparison between HAWK and more conventional approaches, were the SNP calls used in the conventional approach based on the same data that HAWK had access to? If not, it's not really a fair comparison. Also, add in what happens if you use dosage calls.

10) Include explanation of all the ways HAWK might lead to false positives, how likely they are to occur, and what can be done to find out if they are happening.

11) HAWK appears only applicable to case-control, and not quantitative traits. This should be mentioned in the Abstract, and also more directly in the Introduction. The authors should explain how HAWK could be applied to quantitative traits (and ideally demonstrate it).

12) Define the scope where HAWK is to be applied more carefully than "sequencing data". Please provide empirical support using real data for any claims (so, if it applies to RNA-Seq data, provide RNA-Seq support). The current results only appear to support whole genome sequencing use cases.

[Editors' note: further revisions were requested prior to acceptance, as described below.]

Thank you for submitting your article "Association Mapping from Sequencing Reads using *k*-mers" for consideration by *eLife*. Your article has been reviewed by three peer reviewers, and the evaluation has been overseen by a Reviewing Editor and Detlef Weigel as the Senior Editor. The following individual involved in review of your submission has agreed to reveal his identity: Casey S Greene (Reviewer #3).

The reviewers have discussed the reviews with one another and the Reviewing Editor has drafted this decision to help you prepare a revised submission.

Essential revisions:

1) The CVD study seems very odd. The authors use BEB or TSI as response variables, and do association, instead of using CVD status (i.e. cases or controls). Since BEB and TSI are obviously different in population ancestry, the resulting *k*-mers would mostly reflect ancestry-specific markers. Indeed, in Table 3, for several variants, the difference between the frequencies of BEB and TSI groups are so large that they unlikely result from their disease effects. The authors show only results from genes already linked to CVD. It's not clear at all if these genes are ranked high in their findings, had they analyzed all genes. As such, it is almost impossible to interpret the results here – whether it's due to population difference or disease association.

2) In the *E. coli*. analysis: this is more like a true association study. Application to *E. coli* instead of human is a bit odd though. Other than that, if one uses traditional procedure of variant calling and association testing, will the same gene(s) be found? The authors need to demonstrate that their method adds something new.

3) "We agree that deviations from the Poisson assumption are possible, and this motivated us to perform the conservative Bonferroni correction and then assemble *k*-mers into larger sequences, providing an approach to discarding false positives".

To appease the concern about the distributional assumption, it is not really sufficient to state that you are using a conservative type 1 error control. First, because that's not really a sensible way to fix the potential problem (albeit if it is the only practical one then I can accept it). Second, because who's to say if Boneferroni (or any method X) is conservative enough to outweigh potential false positive of a wrongly specified model? One would at least have to specifically demonstrate that.

Also, it would be good, as a baseline, to use an expensive negative binomial test to see how much better things would be if you could afford to do that. Otherwise we don't have a good sense of the cost of assuming Poisson (if any). Additionally, if you did negative binomial with a less conservative measure than Bonferroni, we'd have still an even better sense of how bad the assumptions of your approach are. (Which is not to say one shouldn't use it for practical reasons). At the very least, it would be good to see some kind of further justification that Poisson is reasonable (maybe it is in there and I missed it).

4) "We then perform principal components analysis (PCA) on the matrix".

How does this compare to doing a "standard" genetics PCA on the same samples? Which would you expect to be better, and for what reason? It would be nice to see some sort of head-to-head to get a sense of how different they are.

5) "We ﬁt a logistic regression model involving potential confounders and *k*-mers counts".

"involving"? What precisely does this mean? Not clear what you did from this sentence. More importantly, it looks like you are here abandoning the fundamental part of HAWK, the Poisson testing, in this (population confounding) setting, and instead using standard GWAS techniques on the *k*-mer counts instead of on SNPs? If so, this should be expanded on and explained, and the cost in power discussed; if there is no cost in power, then why not use just that method instead of HAWK? In fact, it would be good to see a head-to-head comparison of this logistic regression approach with HAWK, when there is no confounding. This section needs more explicit description of what you are doing, why you are doing it, and what the pros and cons are.

6) "We then ﬁt logistic regression models to predict population identities using ﬁrst two principal components, total number of *k*-mers in each sample and gender of individuals along with *k*-mer counts and obtain ANOVA p-values of the *k*-mer counts using χ^2^-tests"

You need to expand conceptually what is going on here. Why do you need the genders here? And why the total *k*-mer counts? Why not just the PCA? What are you trying to demonstrate exactly? Do you want to show that the PCs by themselves correct for confounding? Are you saying that gender is also a confounder here? If so, why? And if so, how would you account for it or other necessary covariates in your Poisson model? Please expand the thought process here.

7) "We observed that the ﬁrst two principal components separate the two populations and the ﬁrst principal component correlates with sequencing depth indicated by size of the circles in the ﬁgure".

It seems that *only* PC2 is separating the populations; PC1 is most certainly not. Did you remove the mean from your data? PCA is a low rank approximation to the covariance matrix, and thus if you don't remove the mean, you don't really have a covariance matrix, and the first PCA will then just capture the mean, which may be what you are seeing here. When you say with "sequencing depth", specifically what sequencing depth do you mean here? Maybe this is related to the issue of the mean. (If this is hidden to you by using software, then it is trivial do implement from scratch in order to achieve full clarity.)

8) "We then performed simulation experiments to test whether associations can be detected after correcting for confounding factors. […] The fraction of runs association was detected are shown in Appendix 1 Figure 9"

You do this experiment, you have a plot in the appendix, but you fail to tell the reader what the results/take-home message are for it in the main text. Please add this.

9) "It shows that all of the −log_10_(adjusted p-values) are close to zero which is expected since the ﬁrst two principal components separate the populations"

Please draw a QQ plot for the adjusted p-values so we can see just how calibrated thing are. I'd also like to see QQ plots for the distribution of p-values in your simulations in general and on the real data analyses (both for population trait and otherwise).

---

## [Author Response]

Essential revisions:1) Demonstrate that the method is robust to problems of population structure.

We have shown that population stratification can be inferred from *k*-mer data and that population stratification and other confounders can be used to adjust p-values. We have added subsections titled “Detecting population structure” and “Correcting for population stratification and other confounders” in the Materials and methods section and “Detection and correction for confounding factors” in the Results section. These describe our approach. We have also provided some validation results.

2) Statistical problem: to test if k-mers are differentially represented in cases vs. controls, the authors do a simple two-sample Poisson comparison using likelihood ratio test. This makes the implicit assumption that the read counts are Poisson distributed and the Poisson rates are constant in cases and in controls. The latter assumption is generally problematic for read count data, as biological differences can create additional variations of read counts. Please show empirical evidence that the counts are Poisson, and indicate what could go wrong if they are not.It is known in RNA-seq data, for example, that read counts from multiple samples cannot be adequately modeled by a Poisson distribution, and a negative binomial distribution is used to model "overdispersion" (some relevant papers are edgeR (Small-sample estimation of negative binomial dispersion, with applications to SAGE data, Robinson & Smith, Biostatistics, 2008) and WASP (PMID: 26366987)).It is not entirely clear that using negative binomial (NB) would solve the problem in this case, as the underlying Poisson rates may be discrete. For instance, suppose the association signal is driven by a deletion (CNV). Each individual either has the deletion or not. NB makes the assumption that the rate follows a gamma distribution. The authors should show the signal is not driven by overdispersion or confounders. At the minimum, they should have a QQ plot, which shows that for most k-mers, there is no association.

Our rationale behind the Poisson assumption has been explained in the subsection “Finding significant *k*-mers”. This assumption also has a computational advantage, allowing us to compute p-values quickly compared to negative binomial distribution based models which would require an iterative process. We agree that deviations from the Poisson assumption are possible, and this motivated us to perform the conservative Bonferroni correction and then assemble *k*-mers into larger sequences, providing an approach to discarding false positives. We also report fractions of individuals in groups with corresponding *k*-mers present for specific variants highlighted in the paper. Furthermore, the top hits eventually go through another step of adjustment for confounders. We have added the CDF of p-values in the YRI-TSI comparison which shows that for most *k*-mers p-values are not highly significant.

3) A related problem is the possible confounders. For example, if cases and controls have different coverage, and representation of k-mers somehow depends on coverage (GC content is correlated with coverage, for example), one could imagine that this may create some false associations.

To address the problem of potential false associations due to different coverage, we propose to include total number of *k*-mers per sample in our confounder correction step explained in “Correcting for population stratification and other confounders” in Materials and methods and “Detection and correction for confounding factors” in Results.

4) The validation of the method is inadequate. The experiment of classifying population ancestry is a quite artificial – few people will do this kind of analysis. What is much more relevant is application of the method on real sequencing data of some complex phenotype. The authors did do it on a CVD dataset, but only superficially. They applied it only on genes known to be linked to CVD, for example, and it's unclear how the results compare with the alternative approach. To really demonstrate the benefits of HAWK, they should apply it to the full data, and evaluate the results more deeply. How do the results compare with standard GWAS? Any new associations? Can these new associations by validated? Do we learn any biology?

The objective of the association mapping with population identity as the phenotype was to check whether associations of diverse types of variants can be captured using *k*-mers in a setting not confounded by population structure. We have subsequently proposed an approach to correct for population stratification and other potential confounders and applied it to data to map ampicillin resistance in *E. coli* and updated the manuscript. We find hits to a gene known to confer the said resistance.

5) The authors state that Cortex (Iqbal et al.) is not suited to a large number of individuals (Introduction, third paragraph). Please explain why this is the case. A comparison with HAWK would help to motivate their approach.

We have provided an explanation in the third paragraph of the Introduction.

6) Iqbal et al. say that "Current assembly methods also typically ignore pre-existing information, such as a reference sequence or known variants. Although variant discovery should not be biased by such information, neither should this information be discarded." The authors should comment on this with respect to their approach.

A major motivation of our approach is to allow association mapping in organisms with no or incomplete reference genome and not be biased by reference genome in variant calling. An intriguing question is at what stage we can better utilize reference genomes? One answer may be when assembling *k*-mers. We have commented on this in the sixth paragraph of the Introduction.

7) For the comparison between HAWK and traditional genotype GWAS (subsection “Verification with 1000 genomes data”), restrict the analysis to only SNPs, i.e. to variants that a normal GWAS is meant to capture. Then we can see the power difference afforded by Poisson and access to counts vs genotype calls. Also, include the use of dosages, a softer version of a hard variant calls. If the false positive rate is well-controlled here (provide quantile plots of p-values), and you are well-powered, then the much-harder-to-verify claims about finding structural variants are better supported.

A limitation of our approach is it doesn’t allow us to consider only one type of variant. While a single SNP can lead to sequences of maximum length of 61, there are often multiple SNPs nearby making restricting analysis to SNPs difficult. Presence of *k*-mers due to structural variants and contaminants make a straightforward comparison between the approaches hard. Hence we resorted to simulation. We stress that the major motivation of our approach is to allow association mapping without a reference genome. The power issue was discussed to partly explain the larger number hits found by *k*-mer based approach.

8) The authors should discuss the error rate they are interested in and why, and then explain what an appropriate error measure would be. For example, it seems that Storey et al.'s q-values may be more appropriate (Bonferroni is overly conservative).

We have used the conservative Bonferroni correction specifically because it is conservative and thus can ameliorate deviation from the Poisson assumption. We have added an explanation in “Finding significant *k*-mers”.

9) In the comparison between HAWK and more conventional approaches, were the SNP calls used in the conventional approach based on the same data that HAWK had access to? If not, it's not really a fair comparison. Also, add in what happens if you use dosage calls.

Yes, we used the same set of reads used by the 1000 genomes project to call variants.

10) Include explanation of all the ways HAWK might lead to false positives, how likely they are to occur, and what can be done to find out if they are happening.

We have listed possible reasons behind false positives in the subsection “Correcting for population stratification and other confounders” and discussed an approach to deal with them.

11) HAWK appears only applicable to case-control, and not quantitative traits. This should be mentioned in the Abstract, and also more directly in the Introduction. The authors should explain how HAWK could be applied to quantitative traits (and ideally demonstrate it).

We have mentioned it in the Abstract and the Introduction. We have mentioned how we may extend this to quantitative phenotypes at the ends of the subsections “Finding significant *k*-mers” and “Correcting for population stratification and other confounders”.

12) Define the scope where HAWK is to be applied more carefully than "sequencing data". Please provide empirical support using real data for any claims (so, if it applies to RNA-Seq data, provide RNA-Seq support). The current results only appear to support whole genome sequencing use cases.

We have applied our method to whole-genome sequencing data so far. The manuscript has been updated to reflect that.

[Editors' note: further revisions were requested prior to acceptance, as described below.]

Essential revisions:1) The CVD study seems very odd. The authors use BEB or TSI as response variables, and do association, instead of using CVD status (i.e. cases or controls). Since BEB and TSI are obviously different in population ancestry, the resulting k-mers would mostly reflect ancestry-specific markers. Indeed, in Table 3, for several variants, the difference between the frequencies of BEB and TSI groups are so large that they unlikely result from their disease effects. The authors show only results from genes already linked to CVD. It's not clear at all if these genes are ranked high in their findings, had they analyzed all genes. As such, it is almost impossible to interpret the results here – whether it's due to population difference or disease association.

Since the sites were obtained using population identity and not CVD status, they are due to population difference. We merely wanted to explore whether any of those could potentially contribute to higher mortality from CVDs in BEB. The listed sites are not the highest ranked ones and are included as they are in genes known to be linked to CVDs. We have provided clarification in the second paragraph of the subsection “Differential prevalence of variants in genes linked to CVDs in BEB-TSI comparison”. We feel that insights gained from such unconventional methods may help researchers to narrow down on specific variants and study their effects.

2) In the E. coli. analysis: this is more like a true association study. Application to E. coli instead of human is a bit odd though. Other than that, if one uses traditional procedure of variant calling and association testing, will the same gene(s) be found? The authors need to demonstrate that their method adds something new.

We have performed a conventional association study, presented the results in Figure 14 in the appendix and discussed our findings and limitations of such approaches in the last paragraph of the subsection “Association mapping ampicillin resistance in *E. coli*”.

3) "We agree that deviations from the Poisson assumption are possible, and this motivated us to perform the conservative Bonferroni correction and then assemble k-mers into larger sequences, providing an approach to discarding false positives".To appease the concern about the distributional assumption, it is not really sufficient to state that you are using a conservative type 1 error control. First, because that's not really a sensible way to fix the potential problem (albeit if it is the only practical one then I can accept it). Second, because who's to say if Boneferroni (or any method X) is conservative enough to outweigh potential false positive of a wrongly specified model? One would at least have to specifically demonstrate that.Also, it would be good, as a baseline, to use an expensive negative binomial test to see how much better things would be if you could afford to do that. Otherwise we don't have a good sense of the cost of assuming Poisson (if any). Additionally, if you did negative binomial with a less conservative measure than Bonferroni, we'd have still an even better sense of how bad the assumptions of your approach are. (Which is not to say one shouldn't use it for practical reasons). At the very least, it would be good to see some kind of further justification that Poisson is reasonable (maybe it is in there and I missed it).

We have presented our reasoning behind the Poisson assumption in the first paragraph of the subsection “Finding significant *k*-mers”. Earlier we had meant to say, albeit not quantitatively, that when there is deviation from the Poisson assumption, a series of measures is likely to reduce the number of false positives.

We have performed a simulation where data is simulated from negative binomial distributions and p-values were then calculated using likelihood ratio tests for both Poisson and negative binomial distributions. The results are in Appendix 1 Figure 1 along with p-values calculated in the same way on some real data. The results do not indicate that p-values calculated assuming Poisson distribution is notably lower than those found using negative binomial distributions.

4) "We then perform principal components analysis (PCA) on the matrix".How does this compare to doing a "standard" genetics PCA on the same samples? Which would you expect to be better, and for what reason? Would be nice to see some sort of head-to-head to get a sense of how different they are.

We have added a PCA plot obtained from variant calls in Appendix 1 Figure 11A and explained why those should be cleaner than PCA done with *k*-mer counts in the first paragraph of the subsection “Detection and correction for confounding factors”.

5) "we ﬁt a logistic regression model involving potential confounders and k-mers counts"."involving"? What precisely does this mean? Not clear what you did from this sentence. More importantly, it looks like you are here abandoning the fundamental part of HAWK, the Poisson testing, in this (population confounding) setting, and instead using standard GWAS techniques on the k-mer counts instead of on SNPs? If so, this should be expanded on and explained, and the cost in power discussed; if there is no cost in power, then why not use just that method instead of HAWK? In fact, it would be good to see a head-to-head comparison of this logistic regression approach with HAWK, when there is no confounding. This section needs more explicit description of what you are doing, why you are doing it, and what the pros and cons are.

We have rewritten this part to clarify things. We actually think the fundamental part of HAWK is the testing of *k*-mers directly. The LRT based on Poisson lets us effectively and quickly test *k*-mers when there are no confounders and when there are confounders, lets us quickly filter *k*-mers out so that more computationally expensive tests can be performed on the rest. We have conducted simulations to assess loss in power and shown results in Appendix 1 Figure 4. We observe that there is cost in power. We leave designing tests taking into account stochasticity in *k*-mer counts which will hopefully improve power as future work. In any case, since we are fitting logistic regressions, it will be difficult to scale up to all the *k*-mers. The section has been expanded (subsection “Correcting for population stratification and other confounders”).

*6) "We then ﬁt logistic regression models to predict population identities using ﬁrst two principal components, total number of k-mers in each sample and gender of individuals along with k-mer counts and obtain ANOVA p-values of the k-mer counts using* χ^2^-tests".You need to expand conceptually what is going on here. Why do you need the genders here? And why the total k-mer counts? Why not just the PCA? What are you trying to demonstrate exactly? Do you want to show that the PCs by themselves correct for confounding? Are you saying that gender is also a confounder here? If so, why? And if so, how would you account for it or other necessary covariates in your Poisson model? Please expand the thought process here.

The genders of individuals are included to prevent false positives from *k*-mers from X and Y chromosomes in case of sex imbalance in cases and controls. Total number of *k*-mers was included in the model as sequencing biases such as the GC content bias are known which may lead to false positives if the cases and controls are sequenced at different sequencing depths. We have provided a detailed description (subsection “Detection and correction for confounding factors”, second paragraph). These were included to illustrate that they could be confounders in some datasets. However, for this dataset first two PCs adequately adjust for confounders (Appendix 1 Figure 11D). So, gender is not a confounder here. The Poisson model is used when there is no significant confounding. When there are we perform a second phase of adjusting p-values using logistic regression.

7) "We observed that the ﬁrst two principal components separate the two populations and the ﬁrst principal component correlates with sequencing depth indicated by size of the circles in the ﬁgure".It seems that only PC2 is separating the populations; PC1 is most certainly not. Did you remove the mean from your data? PCA is a low rank approximation to the covariance matrix, and thus if you don't remove the mean, you don't really have a covariance matrix, and the first PCA will then just capture the mean, which may be what you are seeing here. When you say with "sequencing depth", specifically what sequencing depth do you mean here? Maybe this is related to the issue of the mean. (If this is hidden to you by using software, then it is trivial do implement from scratch in order to achieve full clarity.)

We meant PC1 and PC2 together separates the populations. The mean was indeed removed from the data. The correlation between PC1 and sequencing depth i.e. average number of reads covering a base is possibly due to not getting reads from hard to sequence regions when the depth is low and may be due to other batch effects (sequencing depth is linked here to the sequencing platform used).

8) "We then performed simulation experiments to test whether associations can be detected after correcting for confounding factors. […] The fraction of runs association was detected are shown in Appendix 1 Figure 9".You do this experiment, you have a plot in the appendix, but you fail to tell the reader what the results/take-home message are for it in the main text. Please add this.

We have added a comment to summarize the plot in the last paragraph of the subsection “Detection and correction for confounding factors”.

9) "It shows that all of the −log_10_(adjusted p-values) are close to zero which is expected since the ﬁrst two principal components separate the populations".Please draw a QQ plot for the adjusted p-values so we can see just how calibrated thing are. I'd also like to see QQ plots for the distribution of p-values in your simulations in general and on the real data analyses (both for population trait and otherwise).

We have added QQ plots for adjusted p-values for 1000 genomes (Appendix 1 Figure 10), QQ plots for Poisson LRT p-values for YRI-TSI comparison (Appendix 1 Figure 5), QQ plots for adjusted p-values *E. coli* ampicillin resistance association mapping (Appendix 1 Figure 12) among others.